

Improving Fine Mineral Dust Representation from the Surface to the Column
in GEOS-Chem 14.4.1
Dandan Zhang[1*], Randall V. Martin[1], Xuan Liu[1,2], Aaron van Donkelaar[1], Christopher R. Oxford[1],
Yanshun Li[1], Jun Meng[3], Danny M. Leung[4], Jasper F. Kok[5], Longlei Li[6], Haihui Zhu[1], Jay R. Turner[1], Yu
Yan[1], Michael Brauer[7], Yinon Rudich[8], and Eli Windwer[8]
[1]Department of Energy, Environmental and Chemical Engineering, Washington University in St.
Louis, St. Louis, Missouri 63130, United States
[2]Scripps Institution of Oceanography, University of California San Diego, San Diego, California
92093, United States
[3]Department of Civil and Environmental Engineering, Washington State University, Pullman,
Washington 99163, United States
[4]Atmospheric Chemistry Observations and Modeling Laboratory, National Science Foundation
National Center for Atmospheric Research, Boulder, Colorado 80301, United States
[5]Department of Atmospheric and Oceanic Sciences, University of California Los Angeles, Los
Angeles, California 90095, United States
[6]Department of Earth and Atmospheric Sciences, Cornell University, Ithaca, New York 14853,
United States
[7]School of Population and Public Health, University of British Columbia, Vancouver, British
Columbia V6T 1Z3, Canada
[8]Department of Earth and Planetary Sciences, Weizmann Institute of Science, Rehovot 76100,
Israel
*Correspondence to*: Dandan Zhang (dandan.z@wustl.edu)





**Abstract**
Accurate representation of mineral dust remains a challenge for global air quality or climate
models due to inadequate parametrization of the emission scheme, removal mechanisms, and
size distribution. While various studies have constrained aspects of dust emission fluxes and/or
dust optical depth, surface dust concentrations still vary by factors of 5-10 among models. In this
study, we focus on improving the simulation of fine dust in the GEOS-Chem chemical transport
model, leveraging recent mechanistic understanding of dust source and removal, and reconciling
the size differences between models and ground-based measurements. Specifically, we conduct
sensitivity simulations using GEOS-Chem in its high performance configuration (GCHP) version
14.4.1 to investigate the effects of mechanism or parameter updates. The results are evaluated by
comparisons versus Deep Blue satellite-based aerosol optical depth (AOD) and AErosol RObotic
NETwork (AERONET) ground-based AOD for total column abundance, and versus the Surface
Particulate Matter Network (SPARTAN) for surface $PM_{2.5}$ dust concentrations. Reconciling modelled
geometric diameter versus measured aerodynamic diameter is important for consistent
comparison. The two-fold overestimation of surface fine dust in the standard model is alleviated by
36% without degradation of total column abundance by implementing a new physics-based dust
emission scheme with better spatial distribution. Further reduction by 16% of the overestimation of
surface $PM_{2.5}$ dust is achieved through reducing the mass fraction of emitted fine dust based on the
brittle fragmentation theory, and explicit tracking of three additional fine mineral dust size bins with
updated parametrization for below-cloud scavenging. Overall, these developments reduce the
normalized mean difference against surface fine dust measurements from SPARTAN from 73% to
21%, while retaining comparable skill of total column abundance against satellite and ground-
based AOD.
**1   Introduction**
Mineral dust exerts significant impacts on air quality as the most abundant aerosol component by
mass globally (Kok et al., 2021), on ecosystem health through nutrient transport and deposition
such as phosphorous (Bayon et al., 2024; Swap et al., 1992) and iron (Jickells et al., 2005), and on
climate through its direct scattering and absorbing of radiation and indirect modifications of cloud
properties (Kok et al., 2017; Liao and Seinfeld, 1998; Mahowald et al., 2014). Despite its
importance, accurate representation of mineral dust remains a challenge for global air quality or



climate models due to inadequate parametrization of the emission scheme (Darmenova et al.,
2009; Kok, 2011; Leung et al., 2023), removal mechanisms (Jones et al., 2022; Petroff and Zhang,
2010; Ryu and Min, 2022; Wang et al., 2014b; Zhang and Shao, 2014; Zhang et al., 2001), and size
distribution (Kok et al., 2017; Mahowald et al., 2014). Observational constraints from satellite have
been applied to reduce the large uncertainty of simulated mineral dust and its emissions
(Mytilinaios et al., 2023; Ridley et al., 2016). However, intercomparison projects with various
models still suggest large variability within a factor of 2 for the total column abundance of mineral
dust, with even larger variability in surface concentrations and deposition by factors of 5-10
(Huneeus et al., 2011; Uno et al., 2006; Wu et al., 2020).
In addition to total column observations, ground-level measurements of mineral dust offer another
promising opportunity to understand mechanisms affecting the accuracy of the surface
concentration simulation and the variable performance from the surface to the total column in
intercomparison projects. The Surface PARTiculate mAtter Network (SPARTAN,
https://www.spartan-network.org/, last access: 4 February 2025) is a globally distributed
monitoring network that measures the chemical components of fine particulate matter ($PM_{2.5}$),
including in arid environments (Liu et al., 2024; Snider et al., 2015). These ground-based
measurements of mineral dust in $PM_{2.5}$ offer new data to evaluate, understand, and improve fine
dust simulation in global models.
Dust emissions play a central role in controlling the surface and total column abundance of
mineral dust (Kok et al., 2014; Leung et al., 2023; Tian et al., 2021). The predicted spatial
distribution particularly affects the downwind dust concentrations through long-range transport
and deposition (Prospero, 1999). A new physics-based dust emission scheme (Leung et al., 2023)
includes recent developments in the parametrization of the threshold of friction velocity for dust
mobilization (Martin and Kok, 2018), combined drag partitioning effects due to rocks (Marticorena
and Bergametti, 1995) and vegetation (Pierre et al., 2014a) for a better representation of exerted
surface friction velocity (Leung et al., 2023), and intermittent dust mobilization due to high-
frequency turbulence (Comola et al., 2019). This dust emission scheme has achieved better spatial
correlations of dust column abundance against ground-based and satellite-derived dust optical
depth in the Community Earth System Model version 2 (CESM2) (Leung et al., 2023, 2024).
However, the effects of these new developments of dust emission scheme on the bias against
ground-based measurements of surface fine dust concentrations are less well known and require



further investigation.
The source and removal of dust in the size bins used in dust parametrizations can vary by orders of
magnitude across the broad size range of mineral dust (Kok, 2011; Wang et al., 2014b; Zhang et al.,
2001). Accounting for this size heterogeneity among dust bins could enable better representation
of the global dust cycle. Prior studies have found an underestimation of coarse dust emissions and
an overestimation of fine dust (Kok, 2011; Kok et al., 2017). While various studies have focused on
developing the representation of coarse or super coarse dust (Kok et al., 2017; Meng et al., 2022),
investigation of the effects of different emission size distributions on ambient fine dust are needed
through comparison with in situ fine dust measurements. In addition, the developments and
improvements of parallel computing in air quality or climate models (Eastham et al., 2018; Harris et
al., 2020; Hu et al., 2018; Martin et al., 2022) offer computational capabilities to extend dust size
bins with explicit treatments that could enable better representation of dust, especially over size
ranges with rapid variation in processes. While the parametrization of dry deposition has been
revisited and evaluated against observations (Emerson et al., 2020), below-cloud or washout
scavenging has been generally limited to lumped treatments for fine and coarse aerosols in the
bulk models (Jones et al., 2022; Wang et al., 2011, 2014a). Developments of the size-resolved
parametrization for below-cloud (washout) scavenging (Wang et al., 2014b) are promising to
improve the wet deposition of fine dust, which is especially important in distant downwind regions
due to long-range transport.
In this study, we implement recent developments of a new dust emission scheme with further
refinements including the clay content and wetness in the top soil layer; reducing the dust
emissions over wet, snow and vegetation covered land surfaces; while constraining the global and
regional source with satellite aerosol optical depth (AOD). We revisit the size distribution of emitted
dust, explicitly track mineral dust with geometric diameter less than 2 µm in four size bins, and
update the parametrization for size-resolved washout scavenging. We conduct sensitivity
simulations using the GEOS-Chem chemical transport model in its high performance configuration
(GCHP) to investigate the effects of these developments. We focus on improving the fine dust
representation in GCHP for better agreement from the surface to the column, by comparisons
against ground-level fine dust measurements, and against the ground-based and satellite-retrieved
AOD over dusty regions of the Sahara, the Middle East and Asia.



## 2  Data sources and model description

### 2.1  Data sources

Ground-based AOD measurements are obtained from the Aerosol Robotic Network (AERONET)
Version 3 Level 2 database with improved cloud screening (Giles et al., 2019). We use satellite
retrievals of AOD from the Deep Blue algorithm (Hsu et al., 2019) based on Collection 6.1 of the
Moderate Resolution Imaging Spectroradiometer (MODIS) instrument aboard the satellite
platforms of Terra with local overpass around 10:30 and of Aqua around 13:30, and the Version 2.0
Deep Blue aerosol global product of the Visible Infrared Imaging Radiometer Suite (VIIRS)
instruments aboard the joint NASA/NOAA Suomi National Polar-orbiting Partnership (Suomi NPP)
and NOAA-20 satellites with local overpass around 13:30 (Cao et al., 2014). We choose the Deep
Blue aerosol product due to its optimization for the retrieval of aerosol properties over bright
surfaces, which is typical over arid regions. We average all Deep Blue aerosol products for the year
2018 at a daily basis. Simulated AOD is coincidently sampled with available daily average Deep
Blue AOD.
We use the Version 4.2 Level 3 gridded cloud-free tropospheric aerosol extinction profile product
during daytime and nighttime of the last 15 years (2007–2021) retrieved from the Cloud–Aerosol
Lidar with Orthogonal Polarization (CALIOP) on board the Cloud–Aerosol Lidar Infrared Pathfinder
Satellite Observations (CALIPSO) satellite for climatological aerosol profiles (Young et al., 2018).
We use global ground-based data from the Surface Particulate Matter Network (SPARTAN;
https://www.spartan-network.org/, last access: 4 February 2025) with filter-based $PM_{2.5}$ chemical
composition data (Liu et al., 2024; Snider et al., 2015). Particles with aerodynamic diameter less
than 2.5 µm are collected on Teflon filters using AirPhoton SS5 sampling stations with a sharp-cut
cyclone (SCC) 1.829 that operates at a target flow rate of 5 litter per minute (Lpm) and analyzed for
fine mineral dust concentrations using X-ray Fluorescence (XRF) and a global mineral dust
equation (Equation (A1); Liu et al., 2022) including correction of attenuation effects due to mass
loading. We use data from sites with at least 10 samples for the 5-year (2019–2023) period after the
network began using XRF. The 5-year averaged surface fine dust concentrations from all 26
SPARTAN sites are listed in Table A1.





**2.2  GEOS-Chem chemical transport model**
We use the GEOS-Chem chemical transport model (https://geoschem.github.io/, last access: 4
February 2025) in its high-performance configuration (Eastham et al., 2018) version 14.4.1 ( (The
International GEOS-Chem User Community, 2024)) with improved performance and usability
(Martin et al., 2022). The model is driven by meteorological inputs from GEOS Forward Processing
(GEOS-FP; https://gmao.gsfc.nasa.gov/, last access: 4 February 2025) with resolution
$0.25° \times 0.3125°$ (~25 km) and 72 hybrid sigma-pressure vertical levels up to 0.01 hPa.
GEOS-Chem simulates detailed oxidant-aerosol chemistry in the troposphere and stratosphere,
with gas-phase mechanism of $HO_x$-$NO_x$-$BrO_x$-VOC-$O_3$ chemistry (Bey et al., 2001; Wang et al.,
2021), coupled to aerosol chemistry for sulfate-nitrate-ammonium (SNA) aerosol (Park et al.,
2004), black carbon (BC) (Wang et al., 2014a), and primary and secondary organic aerosol (Pai et
al., 2020), sea salt (Jaeglé et al., 2011), and natural and anthropogenic dust (Fairlie et al., 2007;
Meng et al., 2021; Philip et al., 2017; Zhang et al., 2013). The gas-aerosol partitioning for SNA is
computed by the HETP v1.0 thermodynamic module (Miller et al., 2024). We use the simple,
irreversible, direct yield scheme for secondary organic aerosol production (Pai et al., 2020). The
effects of aerosol on photolysis rates are computed with relative humidity dependent aerosol size
distributions and optical properties with improved parametrization for the effective radii of
inorganic and organic aerosols (Latimer and Martin, 2019; Ridley et al., 2012; Zhu et al., 2023) and
updated optical properties for aspherical mineral dust (Singh et al., 2024).
The standard dry deposition scheme in GEOS-Chem accounts for gravitational settling,
aerodynamic resistance with respect to turbulent transport within the surface layer, and surface
resistance to particle-surface contact due to Brownian diffusion, impaction, and interception with
an observation constrained parametrization (Emerson et al., 2020; Zhang et al., 2001). The
standard wet deposition scheme includes scavenging in convective updrafts, and in-cloud and
below-cloud scavenging from precipitation (Liu et al., 2001; Wang et al., 2011, 2014a).
Emissions for GEOS-Chem are configured using the Harmonized Emissions Component (HEMCO)
module v3.9.1 (Lin et al., 2021). Global anthropogenic emissions are from the Community
Emissions Data System (CEDS) v2 at $0.5° \times 0.5°$ resolution (Feng et al., 2020). Offline emissions of
lightning $NO_x$ (Murray et al., 2012), biogenic VOCs, soil $NO_x$, sea salt (Weng et al., 2020) and mineral
dust (Sections 2.3 and 4.2) at $0.25° \times 0.3125°$ resolution are included to represent emission





processes at the finest available resolution and to enable consistent emission fluxes across model
resolutions. Open fire emissions are from the daily Global Fire Emissions Database (GFED) v4.1s
(Giglio et al., 2013) at $0.25° \times 0.25°$ resolution. Other default emission inventories in GCHP v14.4.1
include volcanic $SO_2$ emissions (Fisher et al., 2011), marine emissions of dimethylsulfide (DMS)
(Breider et al., 2017) at $1° \times 1°$ resolution, and ammonia at $0.25° \times 0.25°$ resolution (Bouwman et
al., 1997; Croft et al., 2016). We conduct GCHP simulations at C48 (~200 km) resolution for the full
year of 2018 following a one-month spin-up.
**2.3   Default dust emission scheme**
The default dust emission scheme in GEOS-Chem (hereafter GC Dust) originally implemented by
Fairlie et al. (2007) is based on the semi-empirical Mineral Dust Entrainment and Deposition
(DEAD) emission scheme (Zender et al., 2003) and the GOCART topographical source function
(Ginoux et al., 2001) updated to a fine resolution of $0.25° \times 0.25°$ (Meng et al., 2021). The total dust
emission flux in kg m$^{-2}$ s$^{-1}$ is calculated based on Zender et al. (2003) and Fairlie et al. (2007):
$$F_d = f_{bare}S\varphi Q_s \tag{1}$$
where $f_{bare}$ is the bare ground fraction as specified in Zender et al. (2003) to reduce dust emissions
over wet, snow and vegetation covered surfaces:
$$f_{bare} = (1 - A_l - A_{wl})(1 - A_{snow})\left(1 - \frac{\text{LAI}}{\text{LAI}_{\text{thr}}}\right) \tag{2}$$
where $A_l$, $A_{wl}$, and $A_{snow}$ is the fraction of land covered by lakes, wetlands, and snow, respectively.
LAI is the leaf area index, and $\text{LAI}_{\text{thr}}$ is the threshold LAI to reduce the bare soil fraction due to
vegetation cover, which is set to 0.3 m$^2$ m$^{-2}$ by default.
$S$ is the GOCART topographical source function (Ginoux et al., 2001) updated at fine resolution of
$0.25° \times 0.25°$ and multiplied by the fraction of bare surface within each grid cell (Meng et al., 2021);
$\varphi$ is the sandblasting efficiency to convert horizontal saltation flux to vertical dust flux (Marticorena
and Bergametti, 1995):
$$\varphi = 10^{13.4 f_{clay} - 4} \tag{3}$$
where $f_{clay}$ is the clay content in the top soil layer and a global constant value of 0.2 is used to





reduce excessive sensitivity of dust emission fluxes to $f_{clay}$ (Zender et al., 2003). $Q_s$ is the
horizontal saltation flux as described in Section A2.

## 2.4 Size distribution of emitted dust

The default size distribution of emitted dust in GEOS-Chem implemented by Zhang et al. (2013) is
based on the Brittle Fragmentation Theory (Kok, 2011) with parameter values optimized using dust
observations from the Interagency Monitoring of Protected Visual Environments (IMPROVE)
ground-based monitoring network in the United States:

$$\frac{dV_d}{d\ln D_d} = \frac{D_d}{c_V}\left[1 + \mathrm{erf}\left(\frac{\ln(D_d/\overline{D_s})}{\sqrt{2}\ln\sigma_s}\right)\right]\exp\left[-\left(\frac{D_d}{\lambda}\right)^3\right] \tag{4}$$

where $V_d$ is the normalized volume for emitted dust aerosols in diameter of $D_d$ in µm; $c_V$ is the
normalization constant to make the integration total of $V_d$ of 1; $\overline{D_s} = 3.4$ µm is the median diameter
of soil particles; $\sigma_s = 3.0$ is the geometric standard deviation of soil particles; $\lambda$ is the side crack
propagation length, whose value is 8 µm in the default particle size distribution (PSD) used in the
GEOS-Chem (GC PSD), and is 12 µm in the Kok PSD (Kok, 2011).
Table 1. The binning of mineral dust in 4-bin and 7-bin simulations using GEOS-Chem. The
geometric diameter range is listed in the bracket adjacent to each size bin in unit of µm.

| 4-bin simulation | 7-bin simulation |
|---|---|
|  | DSTbin1 (0.2–0.36) |
| DST1 (0.2–2.0) | DSTbin2 (0.36–0.6) |
|  | DSTbin3 (0.6–1.2) |
|  | DSTbin4 (1.2–2.0) |
| DST2 (2.0–3.6) | DSTbin5 (2.0–3.6) |
| DST3 (3.6–6.0) | DSTbin6 (3.6–6.0) |
| DST4 (6.0–12.0) | DSTbin7 (6.0–12.0) |


Dust aerosols are conventionally separated into several dust bins to compromise between
accuracy and computational expense (Ginoux et al., 2001; Zender et al., 2003). Table 1
summarizes the binning of mineral dust in 4-bin and 7-bin simulations. In the GEOS-Chem
standard bulk configuration used here, 4 dust size bins are used including DST1 to DST4 covering





geometric diameter of 0.2–12.0 μm (Fairlie et al., 2007). For DST1, 4 sub-bins of 0.2–0.36 μm, 0.36–
0.6 μm, 0.6–1.2 μm, and 1.2–2.0 μm are further separated for heterogeneous chemistry and AOD
calculations, with shared emission, transport and deposition altogether as DST1 (Fairlie et al.,
2007). To improve submicron dust representation, we implement full separation of the 7 dust bins
for coupled physical and chemical processes in GEOS-Chem, as discussed in Section 4.3.2.
**2.5   Reconciling geometric and aerodynamic diameter**
A recent study has emphasized the importance of reconciling the geometric diameter used in
models and the aerodynamic diameter used in ground-based measurements, especially for
mineral dust with higher particle density of ~2500 kg m$^{-3}$ than the standard density of 1000 kg m$^{-3}$
and with aspherical shapes observed in the atmosphere (Huang et al., 2021). We harmonize the
differences between geometric diameter and aerodynamic diameter based on Reid et al. (2003):
$$D_{aer} = D_{geo} \sqrt{\frac{\rho_d}{\chi \rho_0}} \qquad (5)$$

where $D_{aer}$ is the aerodynamic diameter; $D_{geo}$ is the geometric diameter; $\rho_d = 2500$ kg m$^{-3}$ is the
dust density; $\rho_0 = 1000$ kg m$^{-3}$ is the standard spherical particle density; $\chi$ is the dynamic shape
factor calculated by $\chi = \frac{1}{2}\left(F_s^{1/3} + \frac{1}{F_s^{1/3}}\right)$ and $F_s$ is Stokes form factor (Bagheri and Bonadonna,
2016; Huang et al., 2020) which can be calculated by $\text{HWR}(\frac{1}{\text{AR}})^{1.3}$ where $\text{AR} = 1.70 \pm 0.03$ is the
particle length to width ratio, and $\text{HWR} = 0.40 \pm 0.07$ is the particle height to width ratio (Huang et
al., 2021). With this conversion, the aerodynamic diameter of 2.5 μm corresponds to the geometric
diameter of 1.7 μm. The mass fraction of each simulated dust size bin to the total fine dust mass
concentrations can be calculated by the integration of the dust size distribution of Equation (4) with
the $\lambda$ value of 12 μm of the default PSD used in the GEOS-Chem (GC PSD), which is 68% of DST1
with diameter of 0.2–2.0 μm.
In addition to harmonizing different size types used in models and measurements, prior studies
also suggested that the sharpness of size cut-off of different inlets used to collect PM$_{2.5}$ samples
can affect the measured concentrations (Kenny et al., 2000; Peters et al., 2001). To evaluate the
effects, we obtain the dust size distributions of different inlets by multiplying their penetration
efficiencies (Peters et al., 2001) and GC PSD (Equation (4)).



Figure 1 shows the effects of the sharpness of size cut on the size distribution of collected dust
PM$_{2.5}$ samples. All four inlets have a penetration efficiency of near unity for dust with geometric
diameter less than 1.0 μm, which diminishes to 0.5 at a geometric diameter of 1.7 μm and further
diminishes with increasing diameter. The Well Impactor Ninety-Six (WINS) referenced by the
Federal Reference Method (FRM) exhibits the sharpest size cut. The corresponding dust PSD is
sharply attenuated for geometric diameters greater than 1.7 μm. The resultant effects on the mass
fractions of the dust size bin to be included in dust PM$_{2.5}$ are small, with the mass fraction of DST1
ranging from 65–70%. The mass fraction based on SCC 1.829 as used by SPARTAN differs by only
−0.4% from that based on the original GC PSD without inlet penetration correction. In our Base
simulation using the standard version of GEOS-Chem, we calculate surface PM$_{2.5}$ dust as 67.6% of
DST1 to account for both aerodynamic diameter and inlet collection efficiency. Neglect of these
effects would have increased simulated PM$_{2.5}$ dust concentrations by a factor of 2.

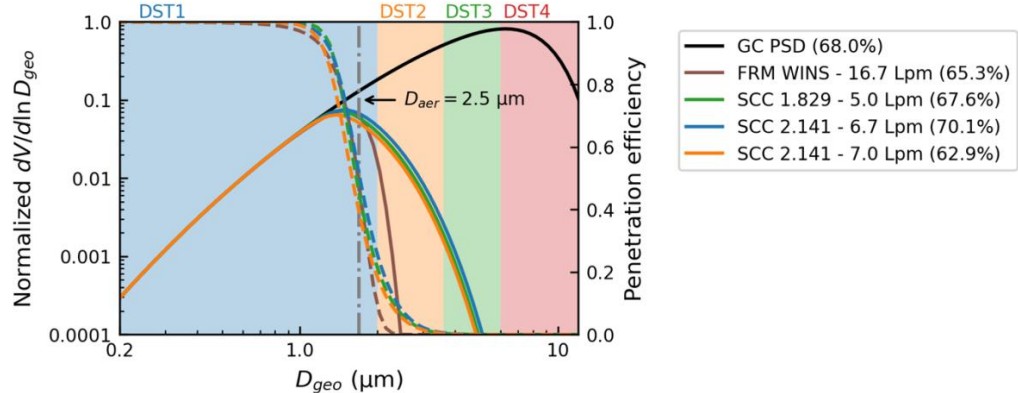


Figure 1. Normalized particle size distribution (PSD) used by default in GEOS-Chem (GC PSD) in
solid black with left axis; penetration efficiencies for different types of PM$_{2.5}$ inlets shown in dashed
colored lines with right axis, including the Well Impactor Ninety-Six (WINS), and three types of
Sharp-Cut Cyclone (SCC) inlets; Solid colored lines show the adjusted GC PSD collected by
different inlets. Grey dash-dotted line indicates the corresponding geometric diameter of 1.7 μm
for the aerodynamic diameter of 2.5 μm. Filled rectangles indicate size ranges of 4 dust size bins.
Percentages adjacent to GC PSD and different inlets are mass fractions of DST1 for the calculation
of PM$_{2.5}$ dust concentrations.



## 3   Strong overestimation of surface fine dust

Figure 2 shows the spatial distributions of the annual total column AOD and surface $PM_{2.5}$ dust from AERONET, SPARTAN, and the Base simulation using the standard version of GEOS-Chem in the year of 2018. Mineral dust largely determines the AOD in AERONET and GEOS-Chem over and downwind of the main dust source regions including the Sahara, Middle East, and the Taklamakan and Gobi deserts in Asia. The simulated AOD over dusty regions (defined here as $AOD_{Dust}/AOD > 0.5$) exhibits a high degree of consistency versus the ground-based observations of AERONET AOD with the regression slope near unity and $R^2$ of 0.7. However, the simulated surface $PM_{2.5}$ dust exhibits a pronounced overestimation by a factor of 2.2 compared to the ground-based measurements of SPARTAN. Simulated $PM_{2.5}$ dust is overestimated at the dusty sites of Abu Dhabi in the United Arab Emirates by 143%, Ilorin in Nigeria by 100%, and Kanpur in India by 75%.

Figure 3 shows the vertical profile of the aerosol extinction normalized by AOD over main dust source regions and associated downwind regions, to understand the significant performance difference between the surface and the column. The simulated vertical profile shows excellent agreement against the 15-year (2007 to 2021) climatological mean extinction vertical profile from the CALIOP, indicating the vertical distribution of mineral dust is not the main driver of the performance discrepancy between the surface and the column.



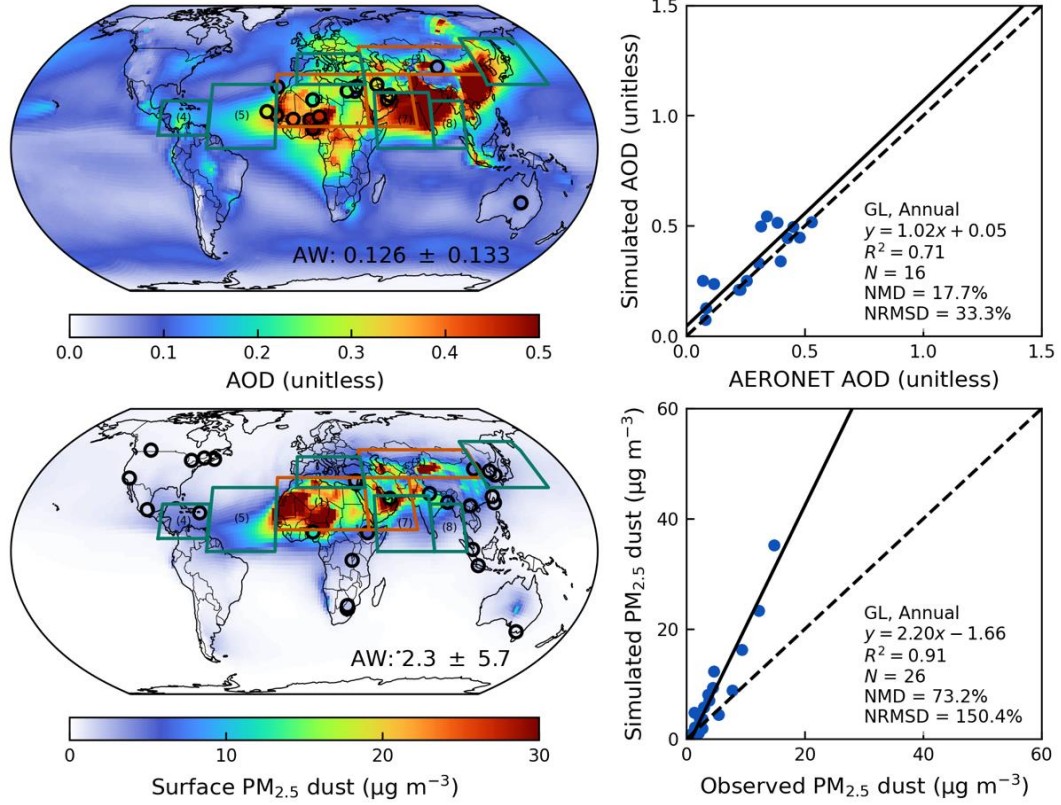

284

Figure 2. Annual simulated aerosol optical depth (AOD) and comparison against ground-based observations from AERONET over dusty regions ($AOD_{Dust}/AOD > 0.5$) (top); Annual simulated surface $PM_{2.5}$ dust and comparison against ground-based measurements from SPARTAN (bottom) from the Base simulation in the year of 2018. Filled circles on the maps represent ground-based observations from SPARTAN and AERONET. Inset values at the bottom right of the maps are area-weighted (AW) mean and standard deviation. Regression statistics including reduced-major-axis linear regression equation, coefficient of variation ($R^2$), total number of points ($N$), normalized mean difference (NMD), and normalized root-mean-square difference (NRMSD) are listed at the bottom right of the scatter plots. Major source regions over land are outlined in red including: 1) the Sahara – SA, 2) Middle East – ME, and 3) Asia – AS. Major dust outflow regions over ocean are outlined in green including: 4) the Caribbean Sea – CRB, 5) the tropical Atlantic Ocean – TAT, 6) the Mediterranean Sea – MED, 7) the Arabian Sean – ARB, 8) the tropical Indian Ocean and the Bay of Bengal – IND, and 9) the northwestern Pacific Ocean – NWP.

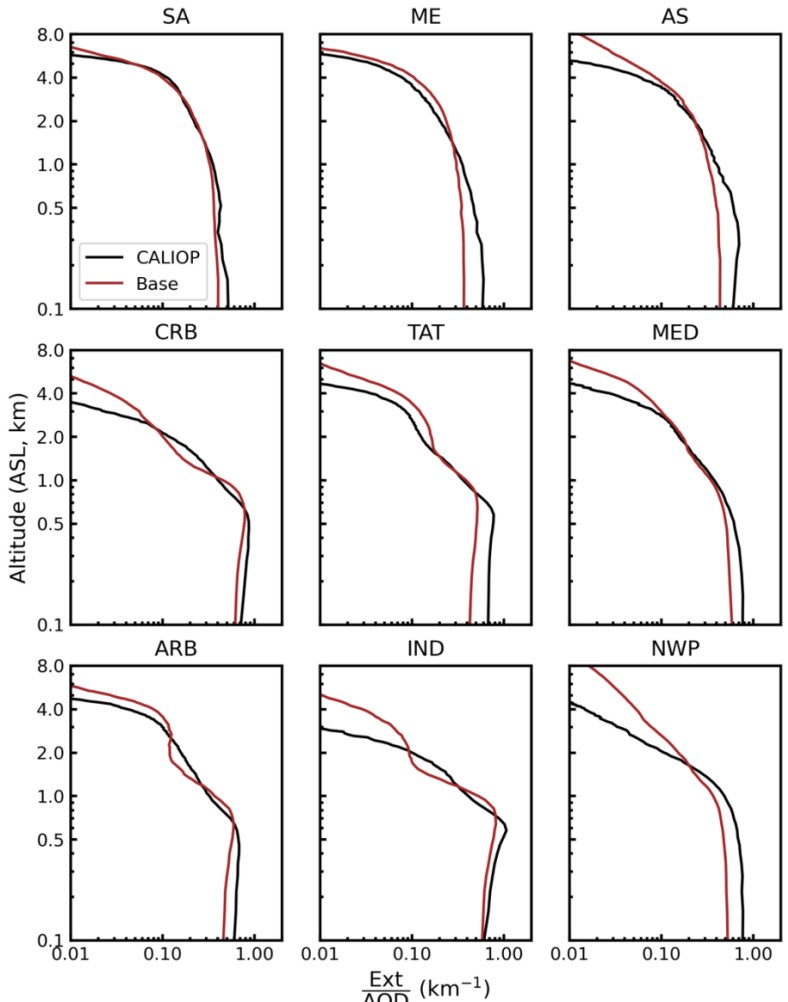

298

Figure 3. Comparisons of the annual extinction vertical profile normalized by total column aerosol optical depth from the Base simulation in the year of 2018 against the 15-year (2007 to 2021) climatological mean extinction vertical profile from the CALIOP over different regions including the major dust source regions over land of the Sahara – SA, Middle East – ME, and Asia – AS, and the major dust outflow regions over ocean of the Caribbean Sea – CRB, the tropical Atlantic Ocean – TAT, the Mediterranean Sea – MED, the Arabian Sea – ARB, the tropical Indian Ocean and the Bay of Bengal – IND, and the northwestern Pacific Ocean – NWP.



## 4 Model revisions to reduce the overestimation of surface fine mineral dust


To reduce the overestimation of surface PM$_{2.5}$ dust, we 1) implement a new dust emission scheme
with further refinements for soil properties including the clay content and soil wetness in the top
soil layer and the threshold of leaf area index, 2) revisit the size distribution of emitted dust, 3)
explicitly track dust with geometric diameter less than 2 μm in four size bins, and 4) update the
parametrization for size-resolved below-cloud scavenging.

### 4.1 Sensitivity simulation setup


Figure 4 summarizes the setup of sensitivity simulations to evaluate the effects of algorithmic
modifications and their performance versus satellite-retrieved AOD and surface dust
measurements. The default dust simulation (Base) in GEOS-Chem as implemented by Fairlie et al.
(2007) uses the DEAD emission scheme (Zender et al., 2003) with a topographical source function
(Ginoux et al., 2001; Meng et al., 2021) for natural dust (GC Dust) with 4 dust size bins for emission,
transport and removal with 7 dust size bins for dust optical depth calculation and heterogeneous
chemistry. To improve the spatial distributions of dust total column abundance, we implement a
new dust emission scheme developed by Leung et al. (2023) (DustL23; Emis). Additional
modifications on top of the original DustL23 emission scheme include 1) reducing the sensitivity of
soil clay content by eliminating the multiplication of the factor of the capped soil clay content $f'_{clay}$
(EmisClay); 2) halving the topmost soil wetness in the layer of 0-5 cm to approximate the soil
wetness in the top 1-2 cm layer which is most pertinent to dust emissions (Darmenova et al., 2009;
Wu et al., 2022) (EmisClayWet); and 3) reducing the threshold of $LAI_{thr}$ from 1.0 m$^2$ m$^{-2}$ to 0.5 m$^2$ m$^-$
$^2$ (EmisClayWetLAI$_{thr}$ or Emis*). To further improve the surface fine dust simulation, we update the
GEOS-Chem particle size distribution (PSD) with the PSD developed by Kok et al. (2011)
(Emis*PSD) with a larger value for the side crack propagation length of $\lambda$ which reduced the mass
fraction of emitted fine dust. The Kok PSD was shown to have excellent agreement versus various
soil size measurements (Kok, 2011), especially for fine dust distributions (González-Flórez et al.,
2023). Lastly, we allow for the four dust bins with geometric diameter less than 2 μm to have
separate emission, transport, and dry and wet deposition while halving anthropogenic dust
emissions from AFCID (Emis*PSD7Bins0.5AD), and with updated below-cloud or washout
scavenging parametrization (Emis*PSD7Bins0.5ADWetDep). Each of these changes is examined
below.



The total global annual source strength for each sensitivity simulation is scaled to achieve unity
slope versus Deep Blue AOD (Figure A1) over major dust source regions. The surface PM$_{2.5}$ dust
concentrations are calculated by accounting for aerodynamic diameter and inlet penetration
efficiency (Section 2.5) as 0.676 DST1 for 4-bin simulations, and DSTbin1 + DSTbin2 + DSTbin3 +
0.546 DSTbin4 for 7-bin simulations. We focus our evaluation on the skill in representing in situ
PM$_{2.5}$ dust concentrations measured by SPARTAN, and in representing the spatial variation in
annual mean AOD. Regression equations are calculated using reduced-major-axis linear
regression.

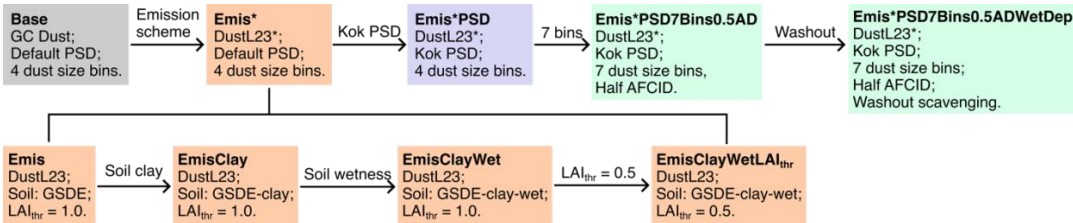


Figure 4. Sensitivity simulation setup. The grey box indicates default settings with the default dust
emission scheme used in GEOS-Chem (GC Dust) with 4 dust size bins (Base). The orange box
indicates the implementation of a modified dust scheme based on DustL23 (Emis*). Modifications
based on the original DustL23 scheme with the soil texture dataset from the Global Soil Dataset for
use in Earth System Models (GSDE) (Emis) include the soil clay content (EmisClay), soil wetness
(EmisClayWet), and threshold leaf area index (EmisClayWetLAI$_{thr}$). The simulation setup for
EmisClayWetLAI$_{thr}$ is the same as that for Emis*. The blue box indicates the modification of size
distribution of emitted dust (Emis*PSD). The green boxes indicate the improvements for fine dust
including explicit tracking of dust with diameter less than 2 μm with a total of 7 dust size bins with
halved anthropogenic fugitive, combustion, and industrial dust (AFCID) emissions
(Emis*PSD7Bins0.5AD), and updating below-cloud (washout) scavenging coefficients
(Emis*PSD7Bins0.5ADWetDep).
**4.2  Improving the spatial distribution of mineral dust with updated emission scheme**
We implement into GEOS-Chem a new physics-based dust emission scheme developed by Leung
et al. (2023) (DustL23) to replace the default dust emission scheme (Section 2.3) used in GEOS-
Chem (GC Dust). The spatial distributions of DustL23 in the Community Earth System Model
version 2 (CESM2) exhibited better correlation against dust optical depth datasets and AERONET





AOD than the DEAD scheme (Leung et al., 2024). We modify DustL23 for implementation into
GEOS-Chem by 1) reducing dust emissions over wet, snow, and vegetation covered surface of
semi-arid regions using Equation (7) below, 2) eliminating the multiplication of the capped clay
content of the topsoil in Equation (8) below, 3) halving the soil wetness in the layer of 0-5 cm to
represent the soil wetness in the top 1-2 cm layer which is most pertinent to dust emissions
(Darmenova et al., 2009; Wu et al., 2022), 4) applying a regional scaling factor of 0.6 over the
Sahara to reduce its emissions (Equation (8)), and 5) scaling the global total emission flux to
achieve unity regression slope versus Deep Blue AOD over dusty regions.
We begin with the formulation for total dust emission flux $F_d$ in kg m$^{-2}$ s$^{-1}$ following Leung et al.

371 (2024):

$$F_d = \eta C_{tune} C_d f_{bare} f'_{clay} \frac{\rho_a \left(u_{*s}^2 - u_{*it}^2\right)}{u_{*st}} \left(\frac{u_{*s}}{u_{*it}}\right)^{\kappa} \text{ for } u_{*s} > u_{*it} \qquad (6)$$

where $\eta$ is an intermittency factor, $C_{tune}$ is a global tuning factor for the emission strength, $C_d$ is the
time-varying soil erodibility coefficient, $f_{bare}$ is the bare ground fraction, $f'_{clay}$ is the clay content in
the topmost soil layer of $f_{clay}$ capped at 0.2, $\rho_a$ is the surface air density in kg m$^{-3}$, $u_{*s}$ is the soil
surface friction velocity in m s$^{-1}$ corrected from the surface friction velocity of $u_*$ by the drag
partitioning effects of $F_{eff}$, $u_{*it}$ is the dynamic or impact threshold friction velocity in m s$^{-1}$, $u_{*st}$ is
the standardized wet fluid threshold friction velocity in m s$^{-1}$, and $\kappa$ is the fragmentation exponent.
Note that we use $u_{*st}$ in the denominator of Equation (6) following Kok et al. (2014) instead of $u_{*it}$
following Leung et al. (2023) for tuning purpose. The parametrization details for these factors
following Leung et al. (2023) can be found in Appendix Section A3.
We modify the DustL23 scheme (Leung et al., 2023) by adopting the equation for the bare ground
fraction in Zender et al. (2003) to reduce dust emissions over wet, snow and vegetation covered
surfaces with the dry erodible land fraction taken from satellite-based land cover:
$$f_{bare} = A_{erod}(1 - A_{snow})\left(1 - \frac{\text{LAI}}{\text{LAI}_{\text{thr}}}\right) \qquad (7)$$

where $A_{erod}$ is the area fraction of erodible surfaces including barren and sparsely vegetated land
cover taken from the MODIS Land Cover Climate Modeling Grid (CMG) (MCD12C1) Version 6.1 data
product; $A_{snow}$ is the area fraction of snow cover, LAI is the leaf area index (Yuan et al., 2011), and



$LAI_{thr}$ is the threshold LAI to reduce the bare soil fraction due to vegetation cover. We set an
intermediate value of $LAI_{thr} = 0.5$ m$^2$ m$^{-2}$ instead of 1.0 m$^2$ m$^{-2}$ in Leung et al. (2023) to represent
the reduction in dust emissions from sparse vegetation over semi-arid regions, which is more
similar to the value of 0.3 used in prior work (Mahowald et al., 1999; Zender et al., 2003).
The enhancement factor $f_m \geq 1$ for the wet fluid threshold friction velocity due to soil wetness is
calculated using Equations (A8) and (A9), but with spatially varying clay content $f_{clay}$ in the top soil
layer. The gridded $f_{clay}$ dataset is taken from the Global Soil Dataset for use in Earth System
Models (GSDE) with various inputs from global and regional soil database (Shangguan et al., 2014),
rather than the machine-learning trained Soil Grids v2.0 dataset with very few observations over
arid regions (Poggio et al., 2021) used in Leung et al. (2023). In addition, we reduce the sensitivity of
dust emissions to clay content by eliminating the multiplication of the capped clay content $f'_{clay}$.
Soil wetness is taken from the parent meteorological inputs of GEOS-FP, targeted at the top 5 cm
layer, and is reduced by half to approximate the soil wetness in the top 1-2 cm layer which is most
pertinent to dust emissions (Darmenova et al., 2009; Wu et al., 2022).
The global scaling factor $C_{tune}$ is determined by the reduced-major-axis linear regression slope of
simulated AOD versus satellite-retrieved AOD over dusty regions ($\frac{AOD_{Dust}}{AOD} > 0.5$) in this study to
constrain the intensity of dust emissions, whose values corresponding to different emission
schemes are listed in Table A2. Additionally, a regional scaling factor of 0.6 over the Sahara ($C_{sah}$)
and unity elsewhere is applied to reduce regionally excessive dust emissions.
The final formulation for dust emission flux is:

$$F_d = \eta C_{sah} C_{tune} C_d f_{bare} \frac{\rho_a \left(u_{*s}^2 - u_{*it}^2\right)}{u_{*st}} \left(\frac{u_{*s}}{u_{*it}}\right)^\kappa \text{ for } u_{*s} > u_{*it} \qquad (8)$$


The calculated offline dust emissions at $0.25° \times 0.3125°$ resolution using Equation (8) are then used
to drive GCHP simulations at C48 resolution. The spatial distributions predicted from different
emission schemes are evaluated against satellite-based Deep Blue AOD, ground-based AERONET
AOD, and SPARTAN surface PM$_{2.5}$ dust measurements.
Figure 5 shows the spatial distributions of annual dust emission fluxes and dust optical depth
predicted from different emission schemes, with Figure 6 showing the comparisons against Deep

Blue satellite AOD globally and over major dust source regions. Comparison of the Base and Emis

schemes reveals that the latter captures more secondary dust emission spots, especially over theSahara, and inland dust sources in Saudi Arabia. However, the comparison against Deep Blue AOD

over the Sahara is degraded versus the default scheme (Figure 6). As suggested by prior studies,

soil clay content is an important factor affecting the threshold friction velocity (Fécan et al., 1999;

Tian et al., 2021; Zender et al., 2003) and sandblasting efficiency (Zender et al., 2003), and is often

tuned for the optimization of dust emissions (Leung et al., 2024; Tian et al., 2021). Eliminating the

multiplication of the capped clay content of $f'_{clay}$ reduces the dust emission sensitivity to the clay

content, increasing emissions from the Bodélé Depression in Chad and El Djouf across the border

of Mauritania and Mali over the Sahara, from the Rub' al Khali desert in the inland Saudi Arabi, and

Taklamakan desert in the northwest China (Figure 5, EmisClay). Correspondingly, the $R^2$ from the

linear regression against Deep Blue AOD is improved from 0.60 to 0.70 over the Sahara, from 0.68

to 0.77 over the Middle East, and from 0.35 to 0.56 over Asia (Figure 6). The other two modifications

of halving soil wetness (EmisClayWet) and setting $LAI_{thr}$ to 0.5 $m^2 m^{-2}$ (EmisClayWetLAI$_{thr}$) slightly

improve the spatial distribution of dust emissions by reducing the underestimation in Asia while

retaining the agreements in the Sahara and Middle East (Figure 6). Using the same dusty region of

the EmisClayWetLAI$_{thr}$ scheme for the comparisons of all dust emission schemes versus Deep Blue

AOD confirms similarly slight improvements of regional dust emissions (Figure A2). Together these

refinements exhibit comparable global performance as the Base simulation versus Deep Blue AOD

with improvements to the relative regional magnitude of dust across the Sahara, Middle East and

Asia.

Figure 7 shows the evaluation of the Emis* (or EmisClayWetLAI$_{thr}$) simulation with ground-based

observations from AERONET and SPARTAN. The overestimation of surface PM$_{2.5}$ dust against the

ground-based measurements of SPARTAN is reduced from 73% (Figure 2) to 37% (Figure 7),

reflecting regional improvements of the spatial distributions especially over the Middle East (Figure

6). The skill in representing AOD in the Emis* simulation remains comparable to that in the Base

simulation shown in Figure 2.

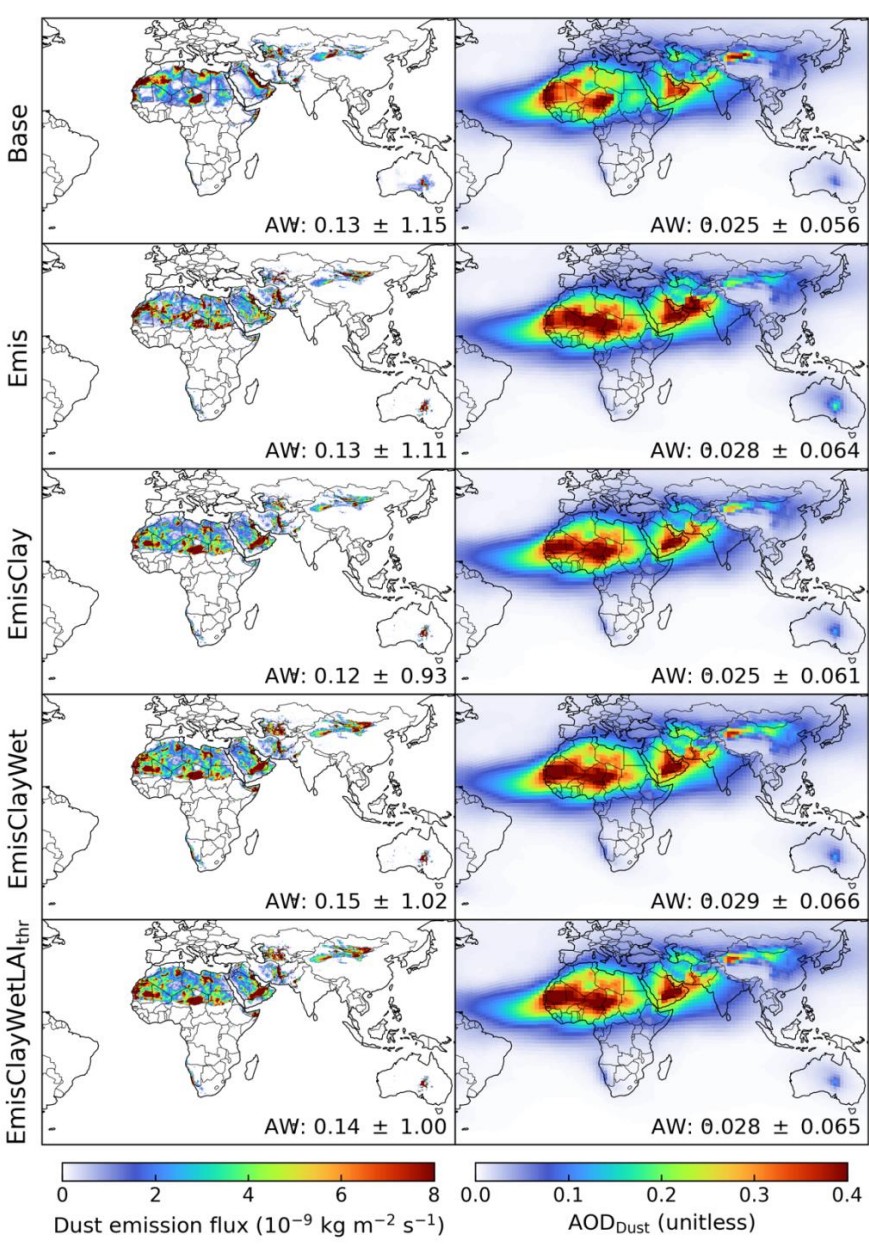

443

Figure 5. Annual dust emission flux (left) and simulated dust optical depth ($AOD_{Dust}$; right) in the
year of 2018 zoomed in over dusty regions of the Sahara, Middle East, and Asia from different
emission schemes as described in Figure 4. Inset values are area-weighted (AW) mean and
standard deviation globally.



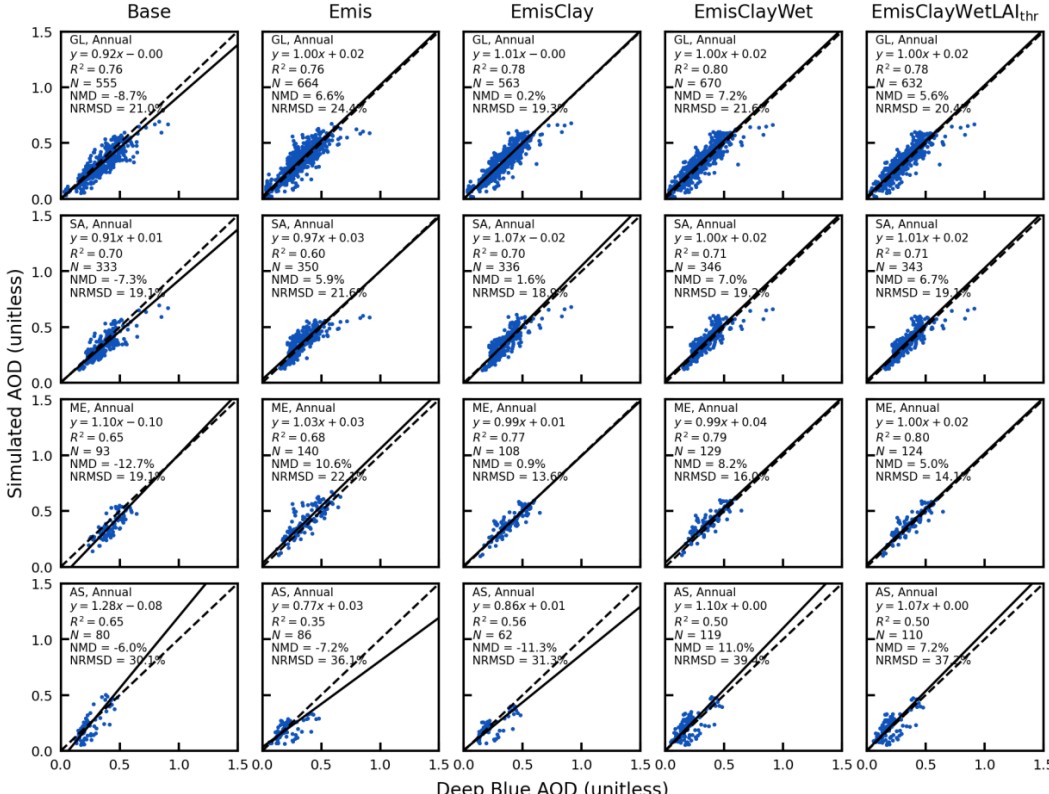

448

Figure 6. Comparisons of annual simulated aerosol optical depth (AOD) versus the Deep Blue
satellite AOD globally (GL) and over main dust source regions of the Sahara – SA, Middle East – ME,
and Asia (AS) with different emission schemes. Regression statistics including reduced-major-axis
linear regression equation, coefficient of variation ($R^2$), total number of points ($N$), normalized
mean difference (NMD), and normalized root-mean-square difference (NRMSD) are in the top left.

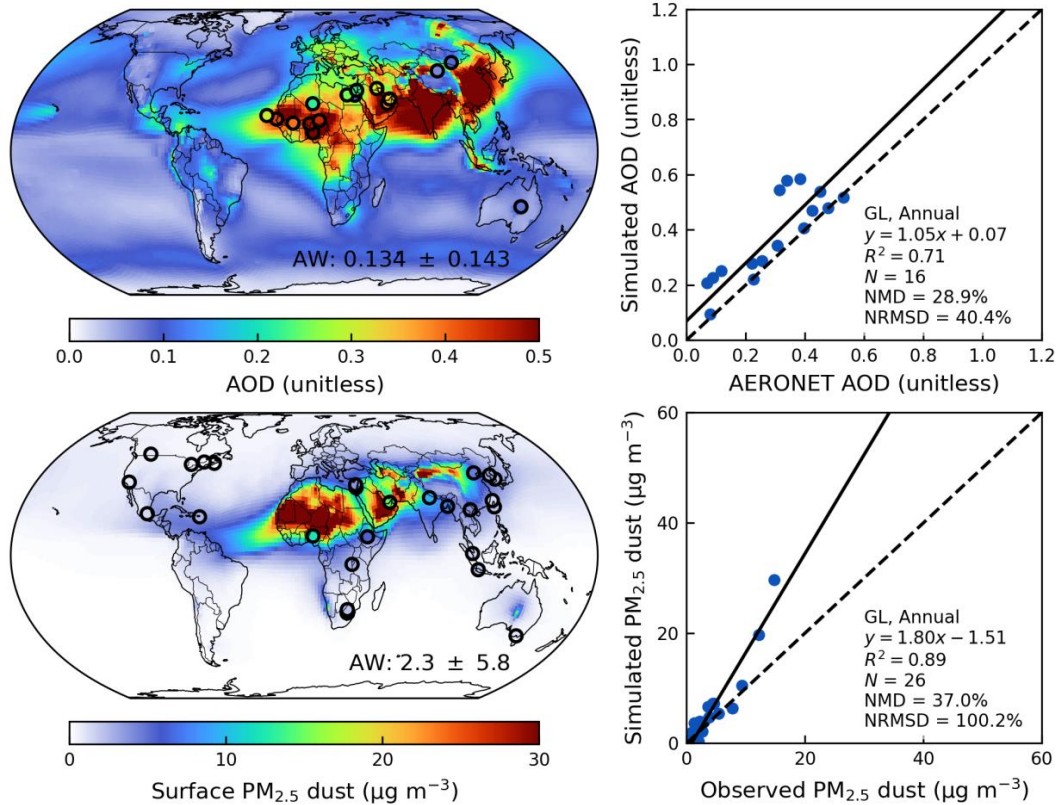

Figure 7. Annual simulated aerosol optical depth (AOD) and comparison against ground-based observations from the AERONET over dusty regions ($AOD_{Dust}/AOD > 0.5$) (top); Annual simulated surface $PM_{2.5}$ dust and comparison against ground-based measurements from the SPARTAN from the Emis* simulation in the year of 2018 (bottom). Filled circles on the maps represent ground-based observations from SPARTAN and AERONET. Inset values at the bottom right of the maps are area-weighted (AW) mean and standard deviation. Regression statistics including the reduced-major-axis linear regression equation, coefficient of variation ($R^2$), total number of points ($N$), normalized mean difference (NMD), and normalized root-mean-square difference (NRMSD) are listed at the bottom right of the scatter plots.

## 4.3 Improving the representation of fine mineral dust

As the size distribution of mineral dust is particularly important for the performance discrepancy between simulated AOD over dusty regions and surface $PM_{2.5}$ dust, we focus on improving its size-



resolved source and sink.

### 4.3.1     Revisiting the size distribution of emitted mineral dust

Figure 8a shows different PSDs including the default PSD used in the GEOS-Chem (GC PSD) based
on the brittle fragmentation theory with the side crack propagation length $\lambda$ of 8 µm (Zhang et al.,
2013), the Kok PSD with $\lambda$ of 12 µm (Kok, 2011), and the Meng PSD focusing on the optimization for
coarse to super coarse dust (Meng et al., 2022), in comparison with the observed PSD from the
2011 Fennec campaign (Ryder et al., 2013). While all modelled PSDs are within the wide range of
PSD from the Fennec campaign, the fraction of emitted DST1 from the Kok PSD exhibits greater
consistency with the Fennec observations than the other two PSDs. Larger discrepancy for the size
distribution with diameter less than ~0.4 µm between the observed PSD from Fennec and
parametrized PSDs is possibly due to anthropogenic aerosol influence (González-Flórez et al.,
2023). In addition, a recent field study in the Moroccan Sahara (González-Flórez et al., 2023)
indicated overall agreement of emitted dust size distributions against the Kok PSD especially at the
fine diameter range. Therefore, we adopt the Kok PSD for the size distribution of emitted mineral
dust in GEOS-Chem. Figure 8b shows the spatial distribution from the Emis*PSD simulation which
remains similar to that from the Emis* simulation in Figure 7. Reduced emissions from DST1 by
using the Kok PSD reduces the overestimation of surface $PM_{2.5}$ dust from 37% to 17% compared to
the ground-based measurements from SPARTAN (Figure 8c).



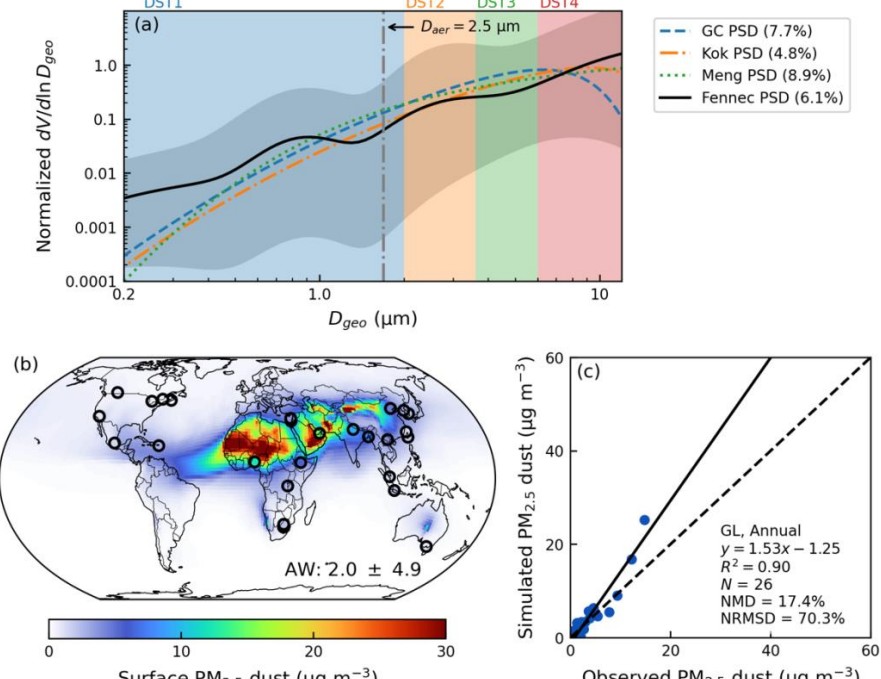


Figure 8. a) Normalized particle size distribution (PSD) of emitted dust based on default PSD used

in GEOS-Chem (GC PSD) (Zhang et al., 2013), the Kok PSD (Kok, 2011), the Meng PSD (Meng et al.,

2022), and the Fennec PSD (Ryder et al., 2013). All PSDs are normalized for a total volumetric

integration of 1 within the diameter range of 0.2 µm to 12 µm used in GEOS-Chem. The grey shades

show the minimum and maximum PSD curves from the Fennec 2011 campaign. Grey dash-dotted

line indicates the corresponding geometric diameter of 1.7 µm for the aerodynamic diameter of 2.5

µm. Filled rectangles indicate size ranges of 4 dust size bins. Percentages adjacent to each PSD are

mass fractions of emitted DST1 over total dust emission flux within diameter range of 0.2 µm to 12

µm. b) Simulated annual surface $PM_{2.5}$ dust from the Emis*PSD simulation in the year of 2018.

Filled circles on the map represent ground-based observations from SPARTAN and AERONET. Inset

values at the bottom right of the maps are area-weighted (AW) mean and standard deviation. c)

Comparison of simulated $PM_{2.5}$ dust versus observed fine dust from SPARTAN. Regression

statistics including the reduced-major-axis linear regression equation, coefficient of variation ($R^2$),

total number of points ($N$), normalized mean difference (NMD), and normalized root-mean-square

difference (NRMSD) are listed at the bottom right.





4.3.2  Improving the size-resolved dry and wet deposition of mineral dust
The default below-cloud (washout) scavenging of dust by rain and snow in GEOS-Chem is
separated for fine (DST1) and coarse dust (DST2 to DST4) (Wang et al., 2011). However, washout
scavenging coefficients strongly depend on aerosol size, varying by 3 orders of magnitude for
diameter ranging from 1 to 10 μm (Wang et al., 2014b). To improve the size-dependent washout
treatment of dust, we update washout rates by rain and snow for 7 dust size bins by (Wang et al.,
2014b):
$$\Lambda = A(D_d)(\frac{P_d}{f_r})^{B(D_d)} \tag{9}$$

where $\Lambda$ is the washout scavenging coefficient in s$^{-1}$ by either rain or snow; $P_d$ is the precipitation
rate in mm h$^{-1}$ falling form upper layers; $f_r$ is the area fraction of precipitation within each grid box;
$A$ and $B$ are empirical constants dependent on dust size $D_d$. Using the same equations for $A$ and $B$
as Wang et al. (2014b), the updated values for different dust size bins are summarized in Table 2.
Table 2. Values of $A$ and $B$ for washout parametrizations by rain and snow for different dust size
bins.

| Diameter (μm) | Rain ($T \geq 268$ K) | | Snow ($248$ K $\leq T < 268$ K) | |
|---|---|---|---|---|
| | $A$ | $B$ | $A$ | $B$ |
| Bin1 (0.2–0.36) | $4.0 \times 10^{-7}$ | 0.71 | $7.3 \times 10^{-6}$ | 0.57 |
| Bin2 (0.36–0.6) | $4.1 \times 10^{-7}$ | 0.71 | $1.3 \times 10^{-5}$ | 0.56 |
| Bin3 (0.6–1.2) | $4.8 \times 10^{-7}$ | 0.72 | $2.7 \times 10^{-5}$ | 0.56 |
| Bin4 (1.2–2.0) | $8.4 \times 10^{-7}$ | 0.73 | $6.0 \times 10^{-5}$ | 0.55 |
| Bin5 (2.0–3.6) | $4.8 \times 10^{-5}$ | 0.88 | $4.2 \times 10^{-4}$ | 0.61 |
| Bin6 (3.6–6.0) | $2.2 \times 10^{-4}$ | 0.87 | $1.3 \times 10^{-3}$ | 0.67 |
| Bin7 (6.0–12.0) | $3.4 \times 10^{-4}$ | 0.84 | $2.4 \times 10^{-3}$ | 0.73 |


Figure 9 shows the size-dependent variations of mineral dust dry and wet deposition. The dry
deposition velocity can vary by a factor of 4.9 among Bin1 to Bin4 with the minimum near the
geometric diameter of 0.5 μm. The washout scavenging coefficient can vary by a factor of 2.6
among Bin1 to Bin4 with the minimum near the geometric diameter of 0.4 μm. Given the steep



increasing strength of emitted dust from Bin1 to Bin4 (Figure 8), there is need to explicitly track dust
within DST1. We evaluate these developments by examining their effects on the fractional
contributions of fine dust to total dust.
Figure 10 shows the fractional contributions of fine dust with geometric diameter less than 2 μm to
total dust ($AOD_{FineDust}/AOD_{Dust}$) from the simulations with a total of 7 dust bins for dry deposition
with updated washout scavenging parametrization and their differences. Due to the dominance of
dry deposition over arid dusty regions, the explicit tracking of fine dust dry deposition slightly
reduces $AOD_{FineDust}/AOD_{Dust}$ over major dust source regions. However, the anthropogenic
contributions to fine dust are correspondingly enhanced over urban and industrial regions, leading
to degraded comparison against SPARTAN measurements (Figure A3). We thus scale the AFCID
emissions by half to reduce the excessive contributions from this uncertain source
(Emis*PSD7Bins0.5AD). In addition, accounting for the steep washout scavenging efficiency across
DSTbin5 to DSTbin7 (Figure 9) with updated washout parametrization would induce enhanced
fractional contributions especially for DSTbin5 (Figure A4) and thus relatively reduce fractional
contributions from fine dust with geometric diameter less than 2 μm to total dust
($AOD_{FineDust}/AOD_{Dust}$). Figure 11 shows the overall performance with all revisions from the
simulation of Emis*PSD7Bins0.5ADWetDep. The reduced-major-axis linear regression slope is
further reduced from 1.53 (Figure 8) to 1.44 with comparable values of NMD against SPARTAN
measurements.
Comparisons against other surface dust datasets also show improved or comparable performance
compared to the Base simulation. Figure A5 shows the comparison against ground-observations
over North America. Using the refined new dust emission scheme with the replacement of the size
distribution from the Kok PSD, explicsitly tracking submicron bins for dry deposition, and updating
the washout scavenging parametrization contribute to a comparable extent to reduce the
overestimation over North America from 43% of the Base simulation to 15% of the
Emis*PSD7Bins0.5ADWetDep simulation. Comparisons against surface concentrations and total
deposition of $PM_{10}$ dust (Li et al., 2022b) for the Emis*PSD7Bins0.5ADWetDep simulation are also
comparable with the Base simulation (Figures A6 and A7). Consistent with prior studies (Leung et
al., 2023; Meng et al., 2021), fine-resolution meteorological fields are needed to capture dust
emission hotspots. The simulated total column AOD would be underestimated by 14% compared
to AERONET, and the surface fine dust would be underestimated by 22% compared to SPARTAN if



the dust emissions are calculated with meteorological fields at C48 resolution (Figure A8). Overall
comparisons for the seasonal mean between the Base and the Emis*PSD7Bins0.5ADWetDep
simulations confirm largely reduced overestimation for the surface fine dust against SPARTAN,
while retaining comparable skill for the total column AOD against AERONET (Figures A9 to A12).
Table 3 summarizes the effects of different modifications on the model performance of total
column AOD and surface fine mineral dust in this study. Strong overestimation of surface $PM_{2.5}$
dust concentrations exist in the Base simulation by a factor of 2.2 versus SPARTAN measured dust.
Updating the dust emission scheme with further refinements in the soil properties reduces the
overestimation of surface $PM_{2.5}$ dust by 36%. The surface overestimation by 37% is reduced to 21%
by updating the size distribution of emitted dust, explicitly tracking dust with diameter less than 2
μm in 4 bins, and updating the parametrization of below-cloud scavenging. The comparisons of
simulated AOD versus AERONET and Deep Blue AOD are comparable for all simulations with the
correlation coefficient of 0.8-0.9, and NMDs from −9% to 31%. The emissions between the Base
and Emis* simulations are comparable with the global annual dust emission of ~2000 Tg yr$^{-1}$, which
is within the range of 1000-5000 Tg yr$^{-1}$ from intercomparison projects (Huneeus et al., 2011; Wu et
al., 2020). As the Kok PSD reduces the mass fraction of fine dust, the total emitted mass is
enhanced to ~3000 Tg yr$^{-1}$ with larger contributions from coarse dust.

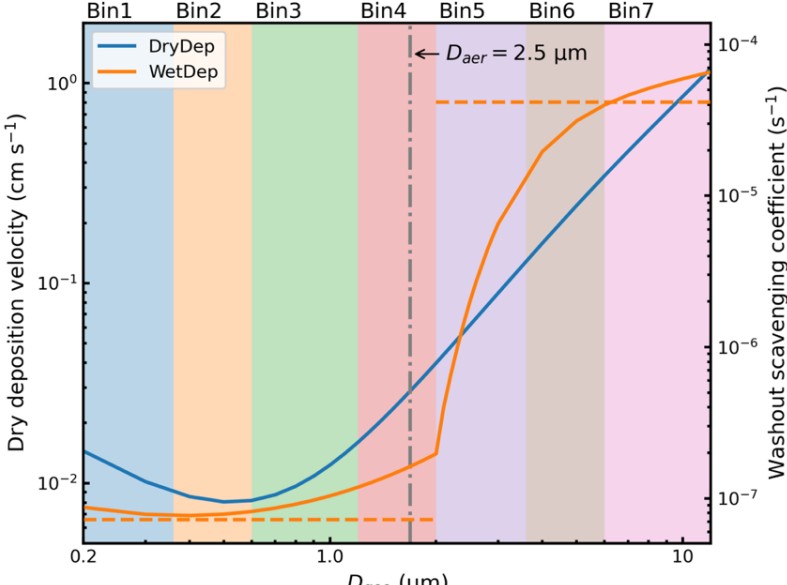


Figure 9. Size-resolved dry deposition velocity over desert (left y-axis) and washout scavenging coefficient by rain (right y-axis). Dry deposition velocity is calculated with the friction velocity of 0.4 m s$^{-1}$ and the particle density of 2500 kg m$^{-3}$ with the default dry deposition scheme used in the GEOS-Chem. Washout scavenging coefficient is calculated with the precipitation rate of 0.1 mm h$^{-1}$ with the updated washout parametrization. Orange horizontal dash lines indicate the default washout scavenging coefficients by rain with the precipitation rate of 0.1 mm h$^{-1}$ for fine aerosol (Bin1 to Bin4) and coarse aerosol (Bin5 to Bin7). Grey dash-dotted line indicates the corresponding geometric diameter of 1.7 μm for the aerodynamic diameter of 2.5 μm. Filled rectangles indicate different simulated dust size bins.

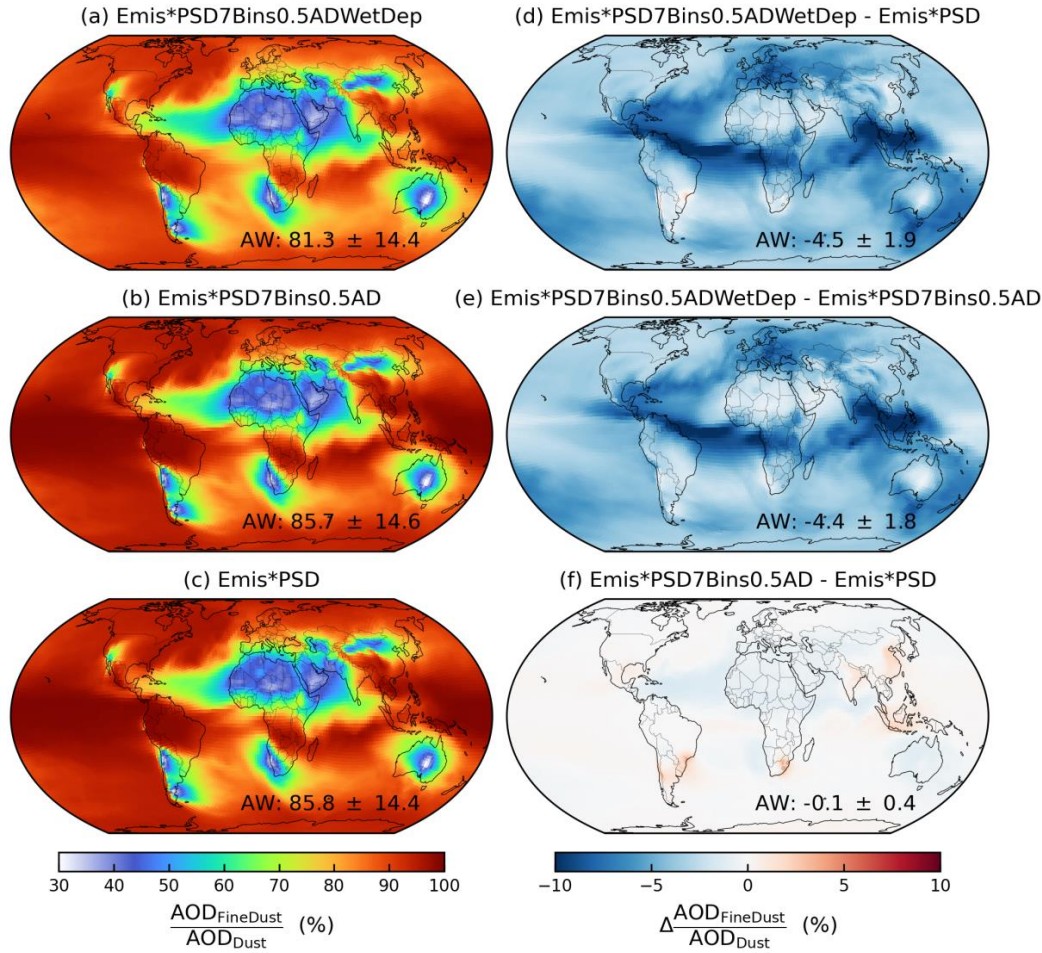

Figure 10. Fractional contributions of fine dust with geometric diameter less than 2 μm to total dust
column abundance ($AOD_{FineDust}/AOD_{Dust}$) from the a) Emis*PSD7Bins0.5ADWetDep, b)
Emis*PSD7Bins0.5AD, c) Emis*PSD and their absolute differences. Inset values at the bottom right
are area-weighted (AW) mean and standard deviation.



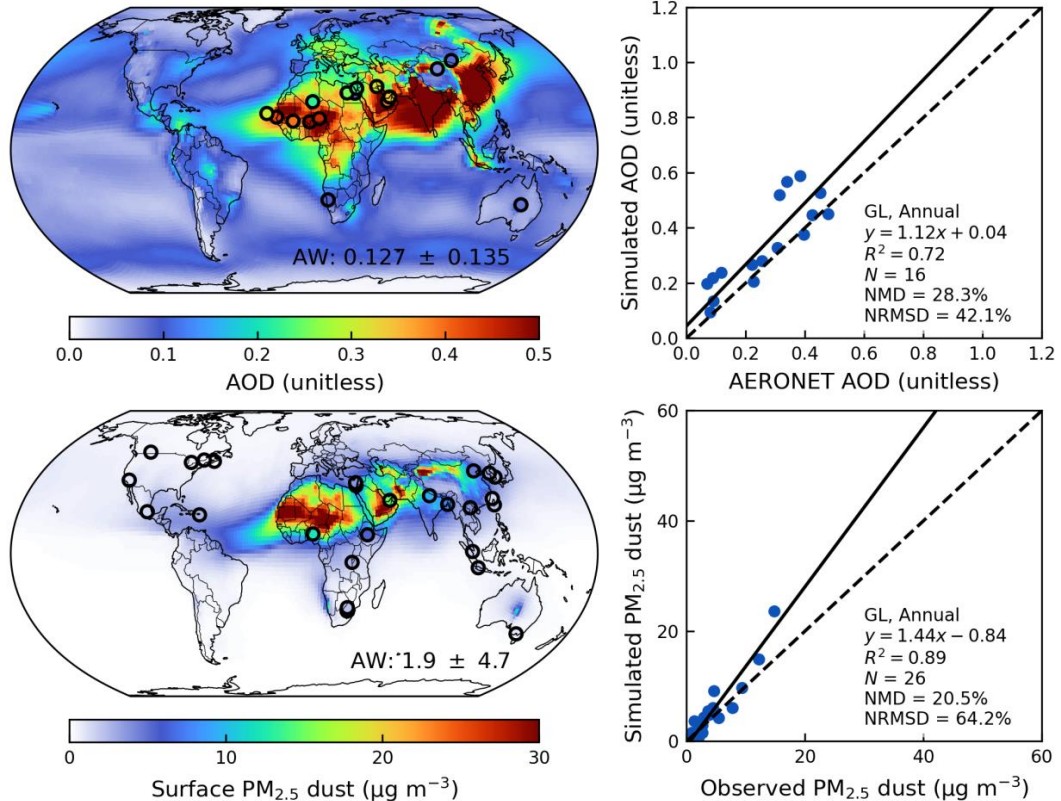


Figure 11. Annual simulated aerosol optical depth (AOD) and comparison against ground-based observations from AERONET over dusty regions ($AOD_{Dust}/AOD > 0.5$) (top); Annual simulated surface $PM_{2.5}$ dust and comparison against ground-based measurements from SPARTAN from the Emis*PSD7Bins0.5ADWetDep simulation in the year of 2018 (bottom). Filled circles on the maps represent ground-based observations from SPARTAN and AERONET. Inset values at the bottom right of the maps are area-weighted (AW) mean and standard deviation. Regression statistics including the reduced-major-axis linear regression equation, $R^2$, total number of points ($N$), normalized mean difference (NMD), and normalized root-mean-square difference (NRMSD) are listed at the bottom right of the scatter plots.

594



Table 3. Effects of different modifications on the model performance of simulated annual surface PM$_{2.5}$ dust versus SPARTAN, and simulated annual aerosol optical depth (AOD) versus AERONET AOD and Deep Blue satellite AOD in terms of the correlation coefficient (r), the reduced-major-axis linear regression slope, and the normalized mean difference (NMD), with associated total annual dust emissions in the year of 2018.

| Simulation | Simulated surface PM$_{2.5}$ dust versus SPARTAN | | | Simulated AOD versus | | | | | | Emissions (Tg yr$^{-1}$) |
| | | | | AERONET AOD | | | Deep Blue AOD | | | |
| | r | slope | NMD (%) | r | slope | NMD (%) | r | slope | NMD (%) | |
| Base | 0.95 | 2.20 | 73.2 | 0.84 | 1.02 | 17.7 | 0.87 | 0.92 | -8.7 | 2025 |
| Emis* | | | | | | | | | | |
| Emis | 0.96 | 1.79 | 47.7 | 0.85 | 1.10 | 26.2 | 0.87 | 1.00 | 6.6 | 2128 |
| EmisClay | 0.95 | 1.69 | 18.1 | 0.86 | 1.05 | 23.7 | 0.88 | 1.01 | 0.2 | 1954 |
| EmisClayWet | 0.94 | 1.84 | 44.3 | 0.87 | 1.11 | 30.7 | 0.89 | 1.00 | 7.2 | 2376 |
| EmisClayWetLAI$_{thr}$ | 0.95 | 1.80 | 37.0 | 0.85 | 1.05 | 28.9 | 0.88 | 1.00 | 5.6 | 2262 |
| Emis*PSD | 0.95 | 1.53 | 17.4 | 0.83 | 1.12 | 29.7 | 0.89 | 1.00 | 4.3 | 3069 |
| Emis*PSD7Bins0.5AD | 0.94 | 1.48 | 25.3 | 0.85 | 1.12 | 28.3 | 0.89 | 1.00 | 3.2 | 2952 |
| Emis*PSD7Bins0.5ADWetDep | 0.95 | 1.44 | 20.5 | 0.83 | 1.11 | 28.7 | 0.89 | 1.00 | 3.6 | 2943 |



## 5  Conclusions

In summary, we evaluate and improve the mineral dust simulation in the GEOS-Chem model by building upon recent ground-based measurements from SPARTAN of mineral dust in PM$_{2.5}$ over land, together with total column AOD from AERONET measurements and from MODIS and VIIRS Deep Blue satellite product. We devote attention to the representation of aerodynamic diameter when comparing with ground-based PM$_{2.5}$ measurements, since representation as geometric diameter in models would introduce two-fold bias. We nonetheless find that the standard GEOS-Chem chemical transport model much better represents columnar AOD with a slope near unity than surface PM$_{2.5}$ dust concentrations which are overestimated by a factor of two. Comparison of simulated extinction profile versus the 15-year climatological CALIOP extinction profile yields overall consistency in the vertical shape (Figure 3), indicating the importance of other dominant factors.

We develop the mineral dust representation in GEOS-Chem with attention to its sources, size distribution, and sinks. We implement a new dust emission scheme based on Leung et al. (2023) with further refinements to the clay content and wetness in the topsoil layer, threshold leaf area index, and reducing dust emissions over snow and vegetation covered land surfaces. The NMD versus surface measurements is reduced by 36% while the simulated AOD better represents the spatial distribution of Deep Blue AOD over dusty regions. To further improve the fine dust representation in GEOS-Chem, we revisit the size distribution of emitted dust and find the Kok particle size distribution (PSD; Kok, 2011) better represents the mass fraction of fine dust measured during the Fennec field campaign over Northern Africa than the default PSD and that its implementation into GEOS-Chem reduces the surface overestimation of PM$_{2.5}$ dust by 20%. We also enable explicit tracking of mineral dust with geometric diameter less than 2 μm in 4 size bins for emission, transport, and deposition with updated parametrization for below-cloud scavenging, which further reduces the overestimation of surface PM$_{2.5}$ dust concentrations to within 21%.

These investigations indicate the importance of size type reconciliation in models versus measurements, the spatial distribution of dust emissions, the size distribution of emitted dust, and the explicit tracking of fine dust bins for more accurate simulation of fine dust abundance from the surface to the column.





**Appendix A: Additional details about dust emission parametrizations, SPARTAN dust, and complementary figures**

**A1. A global dust equation**

We follow a global dust equation for the calculation of surface PM$_{2.5}$ dust concentrations from SPARTAN (Liu et al., 2022):

$$\text{Dust} = [1.89\text{Al} \times (1 + \text{MAL}) + 2.14\text{Si} + 1.40\text{Ca} + 1.36\text{Fe} + 1.67\text{Ti}] \times \text{CF} \quad (A1)$$

where 1.89, 2.14, 1.40, 1.36, and 1.67 are the mass conversion ratios for corresponding mineral oxides; MAL is the mineral-to-aluminum mass ratio of (K$_2$O + MgO + Na$_2$O)/Al$_2$O$_3$; CF is a correction factor (CF) to account for other missing compounds.

**A2. Horizontal saltation flux in standard version of GEOS-Chem**

The default horizontal saltation flux $Q_s$ in GEOS-Chem is based on the parametrization of White (1979):

$$Q_s = C_z \frac{\rho_a}{g} u_{*s}^3 \left(1 - \frac{u_{*ft}}{u_{*s}}\right)\left(1 + \frac{u_{*ft}}{u_{*s}}\right)^2 \text{ for } u_{*s} > u_{*ft} \quad (A2)$$

where $C_z = 2.61$ is the saltation constant; $\rho_a$ is the air density in kg m$^{-3}$; $g = 9.81$ m s$^{-2}$ is the gravitational acceleration; the drag partitioning effects are ignored by default and thus $u_{*s} = u_*$, where $u_*$ is calculated from the wind speed at 10 m $u_{10m}$ based on the logarithmic wind profile within the boundary layer under adiabatic conditions (Marticorena and Bergametti, 1995):

$$u_* = \frac{k u_{10m}}{\ln(z_0/z_{0a})} \quad (A3)$$

where $k = 0.4$ is the von Kármán constant; $u_{10m}$ is the wind speed at 10 m; $z_0 = 10$ m is the reference height; $z_{0a} = 10^{-4}$ m is the surface roughness height. The wet fluid threshold friction velocity of $u_{*ft}$ is the minimum surface friction velocity required to initiate the saltation from the bare soil (Fécan et al., 1999):

$$u_{*ft} = u_{*ft0} \cdot f_m \quad (A4)$$





where $u_{*ft0}$ is the dry fluid threshold friction velocity following Iversen and White (1982):
$$u_{*ft0} = \begin{cases} \dfrac{0.129K}{\sqrt{1.928Re^{0.092} - 1}}, & 0.03 < Re < 10 \\ 0.12K[1 - 0.0858e^{-0.0617(Re-10)}], & Re \geq 10 \end{cases} \tag{A5}$$

where:
$$K = \sqrt{\frac{\rho_p g D_p}{\rho_a}\left(1 + \frac{0.006}{\rho_p g D_p^{2.5}}\right)} \tag{A6}$$

$$Re = 1331 D_p^{1.56} + 0.38 \tag{A7}$$

Where $D_p = 75$ µm is the diameter of soil particle which corresponds to the minimum dry fluid
threshold velocity of $u_{*ft0}$ (Iversen and White, 1982).
The enhancement factor $f_m \geq 1$ is a function of soil wetness (Fécan et al., 1999):
$$f_m = \begin{cases} 1, & w \leq w_t \\ \sqrt{1 + 1.21[100(w - w_t)]^{0.68}}, & w > w_t \end{cases} \tag{A8}$$

where $w$ is the gravimetric soil moisture (kg kg$^{-1}$) in the shallowest soil layer; $w_t$ is the threshold
gravimetric water content above which $u_{*ft}$ increases with soil wetness (Fécan et al., 1999):
$$w_t = 0.01a\left(17 f_{clay} + 14 f_{clay}^2\right) \tag{A9}$$

where $a$ is a tuning factor which is taken as $1/f_{clay} = 5$ by default.
**A3. Additional details about the new dust emission scheme**
The variables used in the calculation for the total dust emission flux $F_d$ (Equation (6)) can be
categorized into meteorological fields including $\eta$, $\rho_a$, and $u_*$, land surface properties including
$f_{bare}$, $f'_{clay}$, $F_{eff}$, and $u_{*it}$, intrinsic soil erodibility properties including $u_{*st}$, $C_d$, and $\kappa$, and a global
tuning factor of $C_{tune}$.
Intermittency effects due to the fluctuation of instantaneous soil friction velocity $\tilde{u}_s$ are reflected in
the intermittency factor of $\eta$, which is denoted by the temporal fraction of active dust emission





ranging from 0 to 1 within a transport time step. The parametrization of $\eta$ is based on Comola et al.

673    (2019):

$$\eta = 1 - P_{ft} + \alpha(P_{ft} - P_{it}) \tag{A10}$$

where $P_{ft}$ and $P_{it}$ are the cumulative probability of instantaneous friction velocity larger than a wet
fluid threshold, and an impact threshold, respectively; $\alpha$ is the fraction of $\tilde{u}_s$ crossing a wet fluid
threshold over the total fraction crossing a wet fluid threshold and an impact threshold.
The calculation of $\eta$ is based on velocity at the saltation height of $z_{sal} = 0.1$ m. Thus the surface
friction velocity of $u_{*s}$, and threshold velocities of $u_{*ft}$ and $u_{*it}$ are first calculated at the saltation
height based on (Marticorena and Bergametti, 1995):

$$u_X(sal) = \frac{u_{*X}}{k} \ln\left(\frac{z_{sal}}{z_{0a}}\right) \tag{A11}$$

where the subscript $X$ can be $ft$, $it$ or $s$, $z_{0a} = 10^{-4}$ m, and $k = 0.386$ is the von Kármán constant.
Assuming a normal distribution of instantaneous soil friction velocity $\tilde{u}_s \sim N(u_s, \sigma^2_{\tilde{u}_s})$, a standard
deviation of instantaneous friction velocity $\sigma_{\tilde{u}_s}$ is a central parameter to calculate the fraction of
active dust emissions within a time step for transportation. $\sigma_{\tilde{u}_s}$ is calculated based on the similarity
theory (Panofsky et al., 1977):

$$\sigma_{\tilde{u}_s} = u_{*s}\left(12 - 0.5\frac{z_i}{L}\right)^{1/3} \tag{A12}$$

where $z_i$ is the planetary boundary layer height, and $L$ is the Monin-bukhov length calculated by
(Panofsky et al., 1977):

$$L = -\frac{\rho_a c_p T u_*^3}{kgH} \tag{A13}$$

where $c_p = 1005$ J kg$^{-1}$K$^{-1}$ is the specific hear capacity of air under constant pressure; $T$ is surface
air temperature; $u_*$ in m s$^{-1}$ is the original surface friction velocity without the drag partitioning
correction; g $= 9.81$ m s$^{-2}$ is the gravitational acceleration; $H$ is the sensible heat flux from
turbulence in W m$^{-2}$.





Given that a normal distribution is assumed, cumulative probabilities of $P_{ft}$ and $P_{it}$ can be
calculated by $P_{ft} = 0.5[1 + \text{erf}\,(\frac{u_{ft}-u_s}{\sqrt{2}\sigma_{\tilde{u}_s}})]$, and $P_{it} = 0.5[1 + \text{erf}\,(\frac{u_{it}-u_s}{\sqrt{2}\sigma_{\tilde{u}_s}})]$. $\alpha$ is the number of crossing
rate of $\tilde{u}_s$ across the wet fluid threshold $C_{ft}$ over the total number of crossing rate of $\tilde{u}_s$ across the
wet fluid threshold $C_{ft}$ and the impact threshold $C_{it}$ (Comola et al., 2019):
$$\alpha = \frac{C_{ft}}{C_{ft} + C_{it}} \tag{A14}$$

The crossing fraction of $\alpha$ is approximated by $\alpha \approx \left[\exp\left(\frac{u_{ft}^2-u_{it}^2-2u_s(u_{ft}-u_{it})}{2\sigma_{\tilde{u}_s}^2}\right) + 1\right]^{-1}$ as suggested by
Comola et al. (2019).
The soil surface friction velocity of $u_{*s}$ is calculated by (Leung et al., 2023; Marticorena and
Bergametti, 1995; Webb et al., 2020):
$$u_{*s} = u_* F_{eff} \tag{A15}$$

where $u_*$ is the surface friction velocity taken directly from the parent meteorological fields; $F_{eff}$ is
the drag partitioning effects due to the presence of non-erodible elements including rocks and
vegetation.
Drag partitioning effects are calculated following Leung et al. (2023):
$$F_{eff} = \left(A_r f_{eff,r}^3 + A_v f_{eff,v}^3\right)^{1/3} \tag{A16}$$

where $A_r$ is the fraction of barren and sparsely vegetated land cover approximated by $A_{erod}$; $A_v$ is
the fraction of short vegetation land cover taken from the MCD12C1 Version 6.1 land cover
product; $f_{eff,r}$ is the drag partitioning effects due to rocks (Marticorena and Bergametti, 1995):
$$f_{eff,r} = 1 - \frac{\ln\left(\frac{z_{0a}}{z_{0s}}\right)}{\ln\left[b_1\left(\frac{X}{z_{0s}}\right)^{b_2}\right]} \tag{A17}$$

where $z_{0a}$ is the aeolian roughness length which the surface roughness of overlaying nonerodable
elements and was taken as the minimum of monthly mean gridded aeolian roughness length
(Prigent et al., 2005); $z_{0s} = \frac{D_p}{15}$ is the smooth roughness length which quantifies the roughness of a



bed of fine soil particles in the absence of roughness elements (Pierre et al., 2014b); $b_1 = 0.7$, $b_2 =$
$0.8$, and $X = 10$ m are empirical constants (Leung et al., 2023). $f_{eff,v}$ is the drag partitioning effects
due to vegetation (Pierre et al., 2014a):
$$f_{eff,v} = \frac{K + f_0 c}{K + c} \qquad (A18)$$

where $f_0 = 0.32$ and $c = 4.8$ are empirical constants (Okin, 2008); $K$ is calculated by $\frac{\pi}{2}\left(\frac{1}{\text{LAI/LAI}_{\text{thr}}} - \right.$
$\left. 1\right)$ (Leung et al., 2023; Okin, 2008).
The wet fluid threshold velocity $u_{*ft}$ is calculated using Equation (A4), except the dry fluid threshold
velocity $u_{*ft0}$ is calculated by (Shao and Lu, 2000):
$$u_{*ft0} = \sqrt{A(\rho_p g D_p + \gamma/D_p)/\rho_a} \qquad (A19)$$

where A = 0.0123 and $\gamma = 1.65 \times 10^{-4}$ kg s$^{-2}$ are empirical constants (Darmenova et al., 2009;
Leung et al., 2023); $D_p = 127 \pm 47$ μm is the median diameter of soil particle as evaluated from
various field measurements in Leung et al. (2023).
Once the saltation is initialized, the threshold velocity required to maintain the saltation
diminishes, which is defined as the dynamic or impact threshold friction velocity $u_{*it}$ in m s$^{-1}$
(Martin and Kok, 2018):
$$u_{*it} = B_{it} u_{*ft0} \qquad (A20)$$

where $B_{it} = 0.82$. A prior study suggested that the impact threshold primarily governed the
saltation flux (Martin and Kok, 2018) and thus $u_{*it}$ is adopted as the governing threshold in Equation

735     (14).

The standardized wet fluid threshold friction velocity $u_{*st}$ was proposed and argued as a central
factor to characterize soil aridity by a prior study (Kok et al., 2014):
$$u_{*st} = u_{*ft}\sqrt{\rho_a/\rho_{a0}} \qquad (A21)$$

where $\rho_{a0} = 1.225$ kg m$^{-3}$ is the standard surface air density.





The fragmentation exponent of $\kappa$ quantifies the sensitivity of $F_d$ to $u_{*s}$ and is capped at 3 to prevent
excessive sensitivity of the model to wind speeds according to (Kok et al., 2014; Leung et al., 2024):

$$\kappa = C_\kappa \frac{(u_{*st} - u_{*st0})}{u_{*st0}} \tag{A22}$$

where $C_\kappa = 2.7 \pm 1.0$ and $u_{*st0} = 0.16 \ \mathrm{m \ s^{-1}}$ are constants.
The time-varying soil erodibility coefficient is a function of $u_{*st}$ only (Kok et al., 2014):

$$C_d = C_{d0} \exp\left(-C_e \frac{u_{*st} - u_{*st0}}{u_{*st0}}\right) \tag{A23}$$

where $C_{d0} = (4.4 \pm 0.5) \times 10^{-5}$ and $C_e = 2.0 \pm 0.3$ are empirical constants.



Table A1. The mean, median, and standard deviation of surface PM$_{2.5}$ dust measured from 26
SPARTAN sites with at least 10 samples in 5 years from 2019 to 2023 globally. Sites are sorted by
the mean surface PM$_{2.5}$ dust concentrations.

| Site | Latitude (°N) | Longitude (°E) | # of samples | Mean (µg m$^{-3}$) | Median (µg m$^{-3}$) | Standard deviation (µg m$^{-3}$) |
|---|---|---|---|---|---|---|
| Abu Dhabi | 24.4 | 54.6 | 136 | 14.8 | 14.1 | 7.4 |
| Ilorin | 8.5 | 4.7 | 58 | 12.2 | 7.1 | 17.1 |
| Kanpur | 26.5 | 80.2 | 18 | 9.3 | 6.2 | 8.2 |
| Dhaka | 23.7 | 90.4 | 53 | 7.7 | 7.4 | 4.1 |
| Addis Ababa | 9.0 | 38.8 | 113 | 5.4 | 5.0 | 1.7 |
| Beijing | 40.0 | 116.3 | 169 | 4.6 | 3.9 | 2.3 |
| Rehovot | 31.9 | 34.8 | 183 | 4.4 | 3.2 | 4.4 |
| Hanoi | 21.0 | 105.8 | 11 | 3.8 | 3.6 | 0.6 |
| Haifa | 32.8 | 35.0 | 141 | 3.6 | 2.5 | 3.7 |
| Seoul | 37.6 | 126.9 | 87 | 2.9 | 2.3 | 1.6 |
| Fajardo | 18.4 | -65.6 | 55 | 2.6 | 1.8 | 2.5 |
| Bujumbura | -3.4 | 29.4 | 15 | 2.6 | 1.9 | 1.4 |
| Kaohsiung | 22.6 | 120.3 | 111 | 2.2 | 2.2 | 0.9 |
| Ulsan | 35.6 | 129.2 | 86 | 2.2 | 1.8 | 1.5 |
| Pretoria | -25.8 | 28.3 | 203 | 2.0 | 2.0 | 0.7 |
| Bandung | -6.9 | 107.6 | 33 | 1.9 | 1.8 | 0.6 |
| Johannesburg | -26.2 | 28.0 | 162 | 1.5 | 1.6 | 0.5 |
| Singapore | 1.3 | 103.8 | 15 | 1.5 | 1.6 | 0.3 |
| Mexico City | 19.3 | -99.2 | 53 | 1.4 | 1.3 | 0.5 |
| Taipei | 25.0 | 121.5 | 204 | 1.3 | 1.0 | 0.9 |
| Pasadena | 34.2 | -118.2 | 220 | 0.9 | 0.8 | 0.3 |
| Lethbridge | 49.7 | -112.9 | 15 | 0.8 | 0.8 | 0.4 |
| Melbourne | -37.8 | 145.0 | 39 | 0.8 | 0.4 | 1.0 |
| Downsview | 43.8 | -79.5 | 22 | 0.6 | 0.6 | 0.2 |
| Sherbrooke | 45.4 | -71.9 | 93 | 0.4 | 0.3 | 0.2 |
| Halifax | 44.6 | -63.6 | 141 | 0.3 | 0.3 | 0.1 |







Table A2. The values of a global tuning factor $C_{tune}$ used for different simulations.

| Simulation | $C_{tune}$ |
| --- | --- |
| Emis* | |
| Emis | $2.358 \times 10^{-2}$ |
| EmisClay | $2.569 \times 10^{-3}$ |
| EmisClayWet | $2.146 \times 10^{-3}$ |
| EmisClayWetLAI$_{thr}$ | $2.170 \times 10^{-3}$ |
| Emis*PSD | $2.945 \times 10^{-3}$ |
| Emis*PSD7Bins0.5AD | $2.892 \times 10^{-3}$ |
| Emis*PSD7Bins0.5ADWetDep | $2.832 \times 10^{-3}$ |


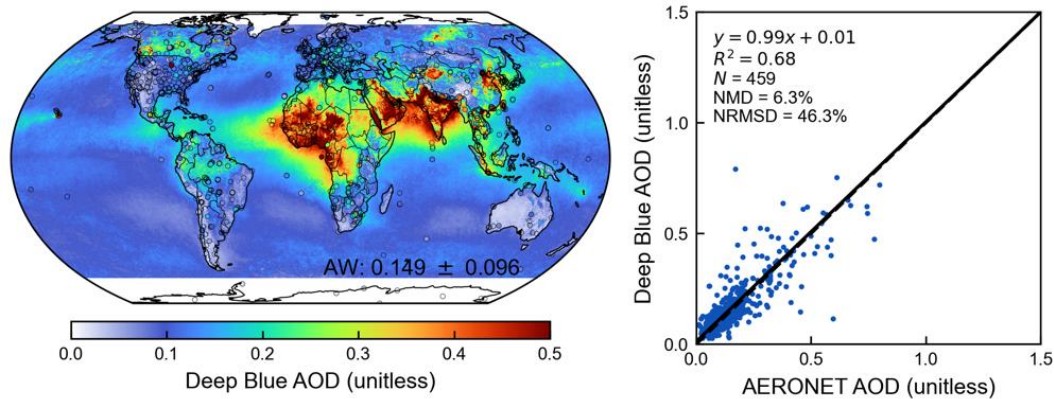


Figure A1. Annual aerosol optical depth (AOD) from the Deep Blue satellite retrieval and
comparison against ground-based observations from AERONET in the year of 2018. Filled circles
on the map represent ground-based observations from AERONET. Inset values at the bottom right
of the map are area-weighted (AW) mean and standard deviation. Regression statistics including
the reduced-major-axis linear regression equation, coefficient of variation ($R^2$), total number of
points ($N$), normalized mean difference (NMD), and normalized root-mean-square difference
(NRMSD) are listed at the top left of the scatter plot.





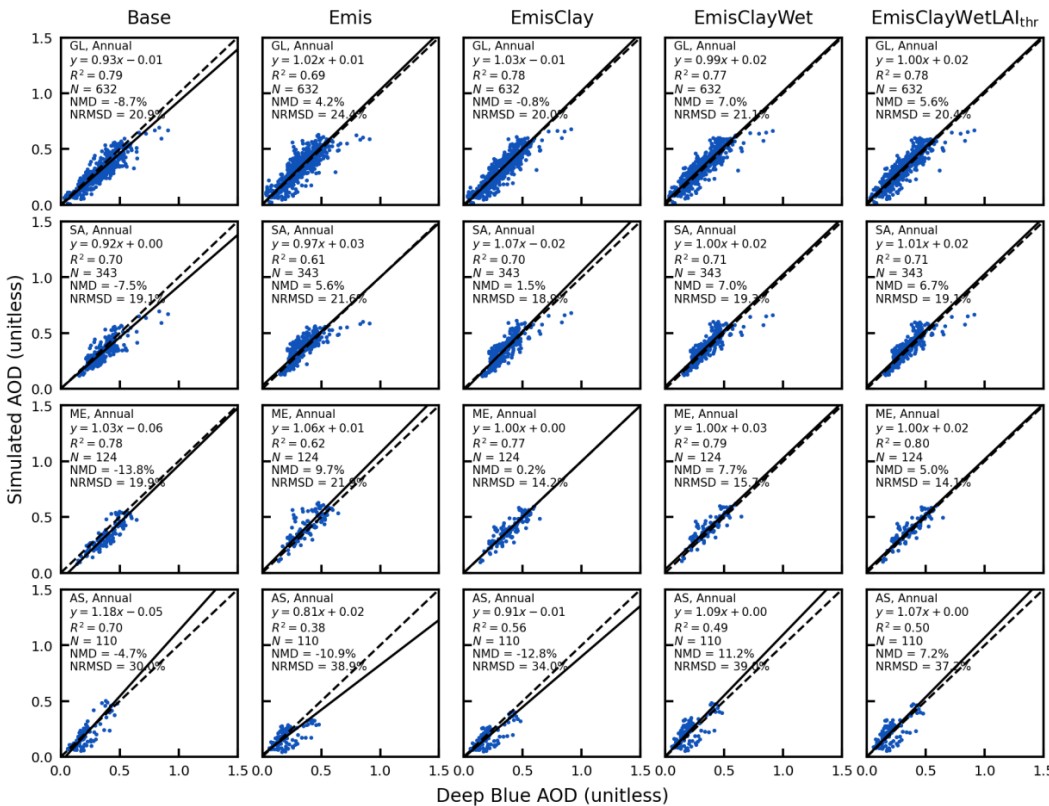


Figure A2. Same as Figure 6 but over the same dust source regions for the EmisClayWetLAI$_{thr}$

scheme for all dust emission scheme comparisons versus Deep Blue AOD.




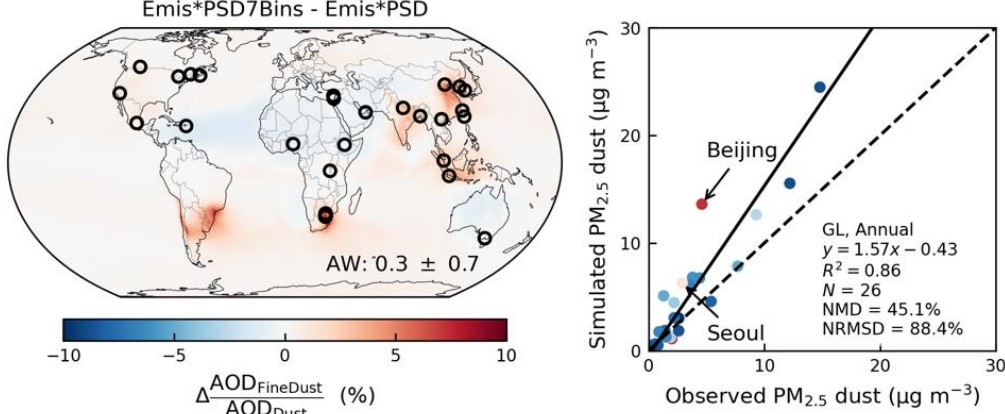


Figure A3. Differences of the fractional contributions of fine dust with geometric diameter less than

2 μm to total dust column abundance ($AOD_{FineDust}/AOD_{Dust}$) between the Emis*PSD7Bins and

Emis*PSD simulations (left); Comparison between simulated $PM_{2.5}$ dust against SPARTAN

measurements from the Emis*PSD7Bins simulation with color coded by the differences of

$AOD_{FineDust}/AOD_{Dust}$ between the Emis*PSD7Bins and Emis*PSD simulations over SPARTAN

sites. Open circles in the map indicate SPARTAN sites. Inset values at the bottom right of the map

are area-weighted (AW) mean and standard deviation. Regression statistics including the reduced-

major-axis linear regression equation, coefficient of variation ($R^2$), total number of points ($N$),

normalized mean difference (NMD), and normalized root-mean-square difference (NRMSD) are

listed at the bottom right of the scatter plot.





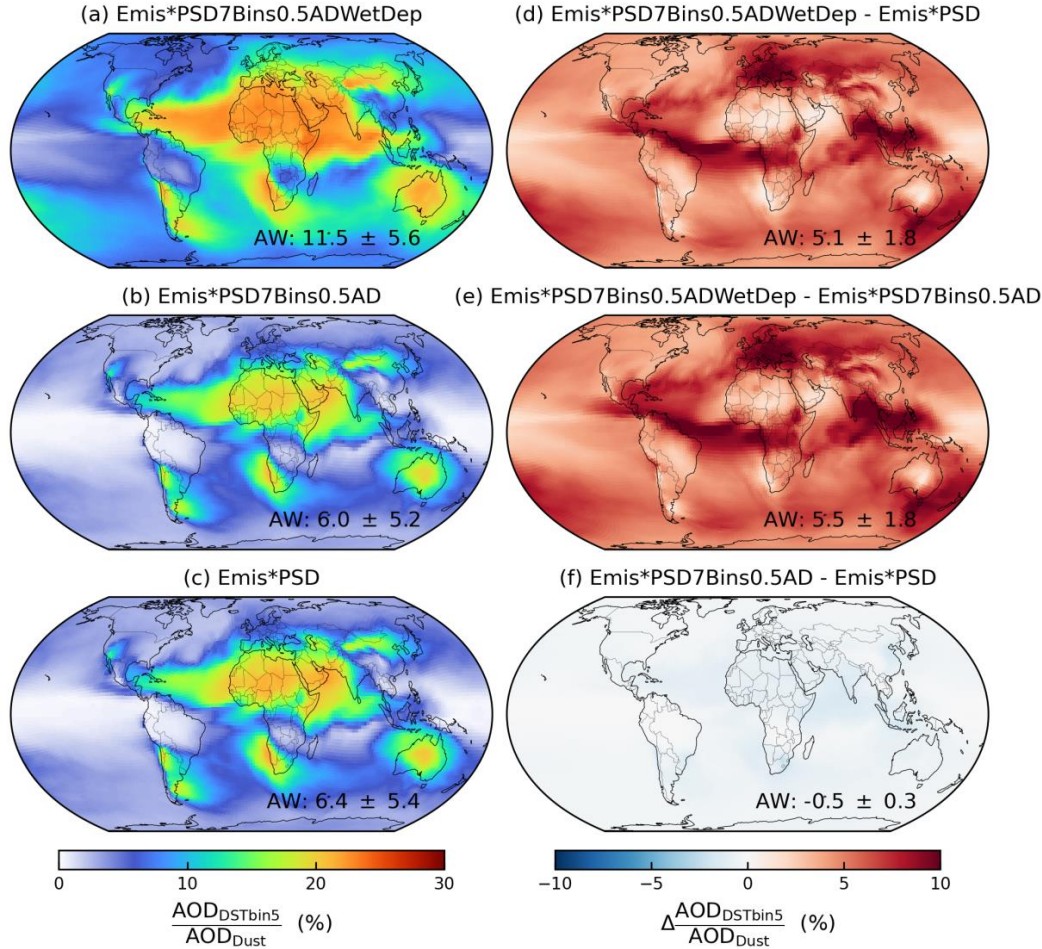


Figure A4. Fractional contributions of DSTbin5 to total dust column abundance
($AOD_{DSTbin5}/AOD_{Dust}$) from the a) Emis*PSD7Bins0.5ADWetDep, b) Emis*PSD7Bins0.5AD, c)
Emis*PSD and their absolute differences. Inset values at the bottom right are area-weighted (AW)
mean and standard deviation.

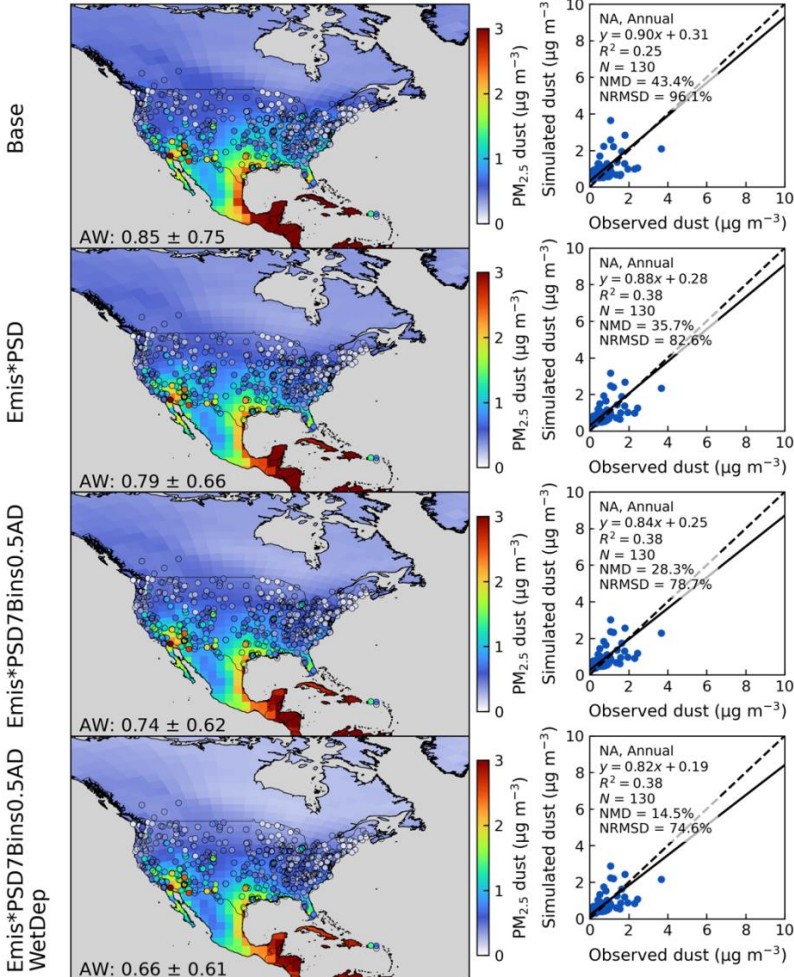

Figure A5. Comparisons of simulated annual surface PM$_{2.5}$ dust against ground-based observations in the year of 2018 over North America from the Base (top), Emis*PSD (second), Emis*PSD7Bins0.5AD (third), and Emis*PSD7Bins0.5ADWetDep (bottom) simulations. Filled circles represent ground-based observations of surface PM$_{2.5}$ dust concentrations. Inset values at the bottom left are area-weighted (AW) mean and standard deviation. Regression statistics including the reduced-major axis linear regression equation, coefficient of variation ($R^2$), total number of points ($N$), normalized mean difference (NMD), and normalized root-mean-square difference (NRMSD) are listed at the top left of right panels.

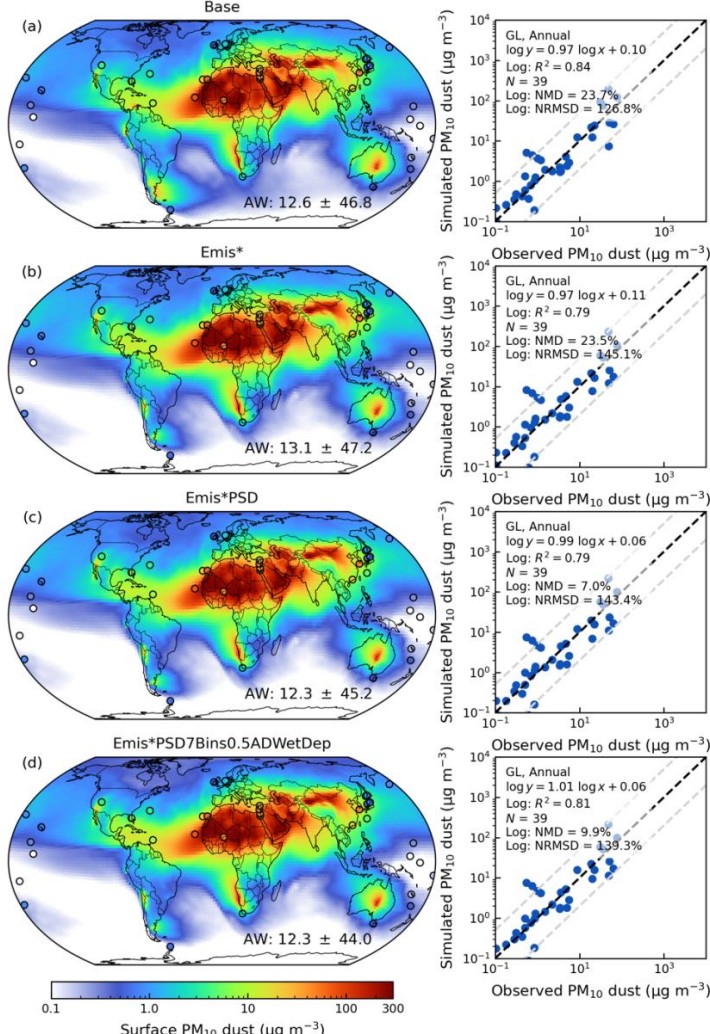

792

Figure A6. Annual simulated surface PM$_{10}$ dust concentrations in the year of 2018 from the

simulations of a) Base, b) Emis*, c) Emis*PSD, and d) Emis*PSD7Bins0.5ADWetDep. Filled circles

represent ground-based observations of surface PM$_{10}$ dust concentrations. Inset values at the

bottom right are area-weighted (AW) mean and standard deviation. Dash lines in the scatter plots

indicate variations within a factor of 5. Regression statistics including the reduced-major-axis

linear regression equation, coefficient of variation ($R^2$), total number of points ($N$), normalized

mean difference (NMD), and normalized root-mean-square difference (NRMSD) are listed at the

top left of right panels.



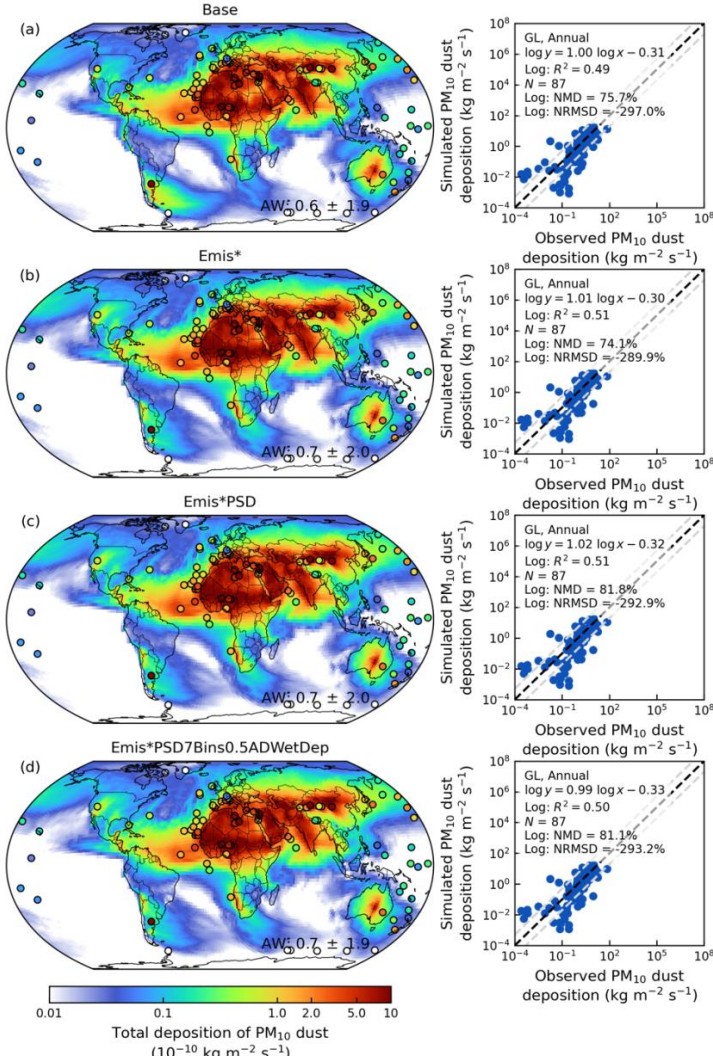

Figure A7. Annual simulated total deposition of $PM_{10}$ dust within the troposphere in the year of 2018 from the simulations of a) Base, b) Emis*, c) Emis*PSD, and d) Emis*PSD7Bins0.5ADWetDep. Filled circles represent ground-based observations of surface $PM_{10}$ dust deposition. Inset values at the bottom right are area-weighted (AW) mean and standard deviation. Dash lines in the scatter plots indicate variations within a factor of 5. Regression statistics including the reduced-major-axis linear regression equation, coefficient of variation ($R^2$), total number of points ($N$), normalized mean difference (NMD), and normalized root-mean-square difference (NRMSD) are listed at the top left of right panels.

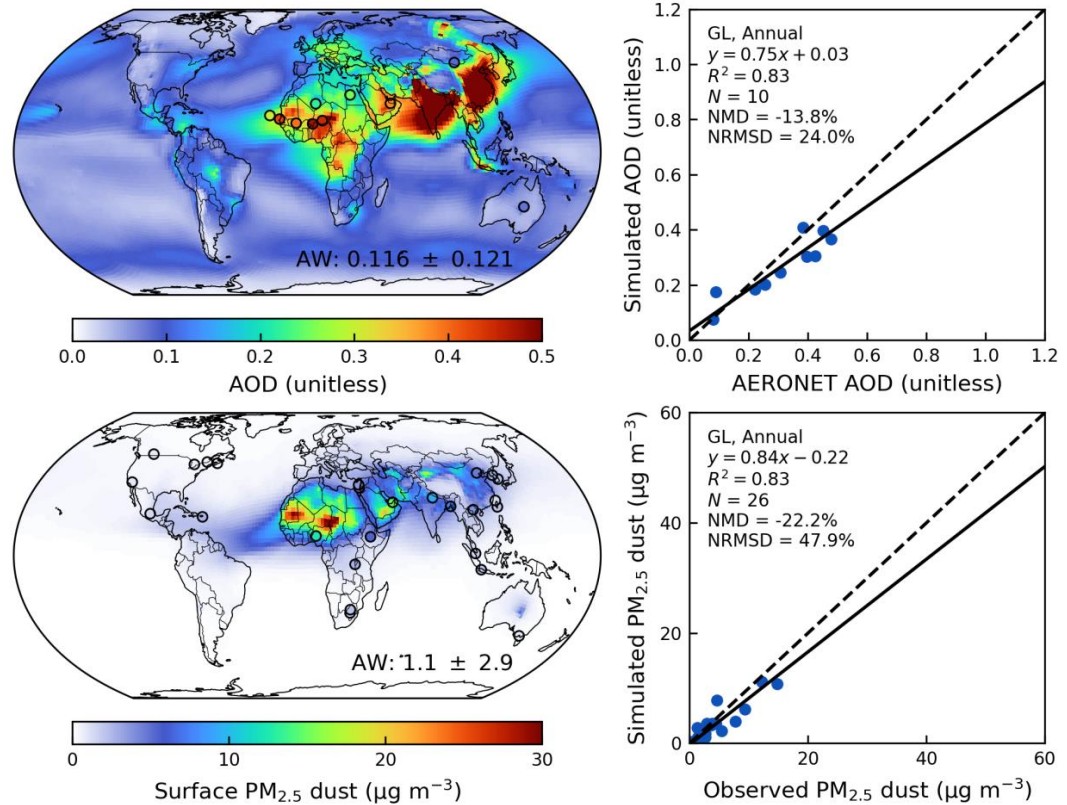

810

Figure A8. Annual simulated aerosol optical depth (AOD) and comparison against ground-based observations from AERONET over dusty regions ($AOD_{Dust}/AOD > 0.5$) (top); Annual simulated surface $PM_{2.5}$ dust and comparison against ground-based measurements from SPARTAN from the Emis*PSD7Bins0.5ADWetDep simulation with the dust emissions calculated at C48 resolution in the year of 2018 (bottom). Filled circles on the maps represent ground-based observations from SPARTAN and AERONET. Inset values at the bottom right of the maps are area-weighted (AW) mean and standard deviation. Regression statistics including the reduced-major-axis linear regression equation, coefficient of variation ($R^2$), total number of points ($N$), normalized mean difference (NMD), and normalized root-mean-square difference (NRMSD) are listed at the top left of the scatter plots.

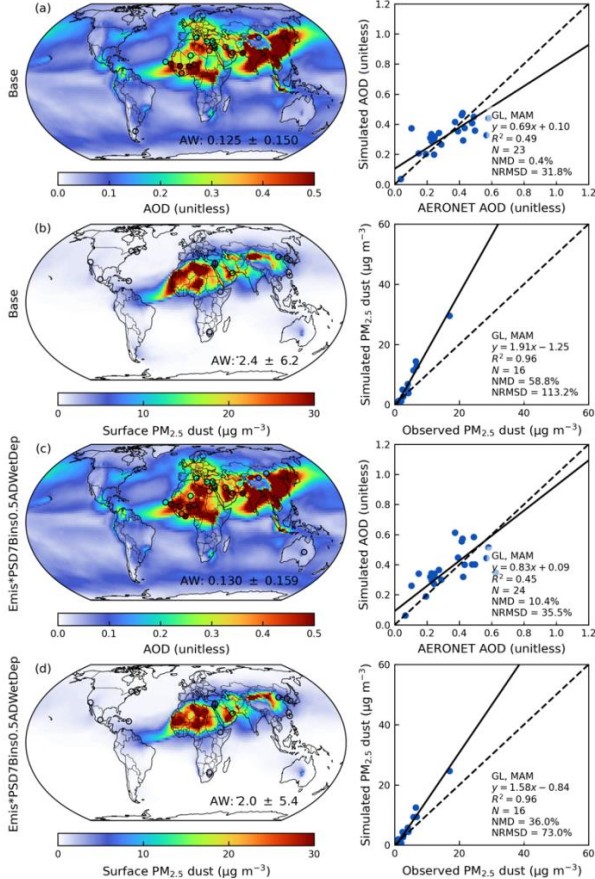

821

Figure A9. Simulated seasonal mean (March, April, and May or MAM) aerosol optical depth (AOD; a
and c) and surface PM$_{2.5}$ dust (b and d) from the Base and Emis*PSD7Bins0.5ADWetDep
simulations. Filled circles on the maps represent ground-based observations from SPARTAN and
AERONET. Inset values at the bottom right of the maps are area-weighted (AW) mean and standard
deviation. Comparisons of simulated AOD versus AERONET AOD over dusty sites ($AOD_{Dust}/AOD >$
0.5), and simulated surface PM$_{2.5}$ dust versus SPARTAN observations are shown in the right panels.
Regression statistics including the reduced-major-axis linear regression equation, coefficient of
variation ($R^2$), total number of points ($N$), normalized mean difference (NMD), and normalized root-
mean-square difference (NRMSD) are listed at the bottom right of the scatter plots.



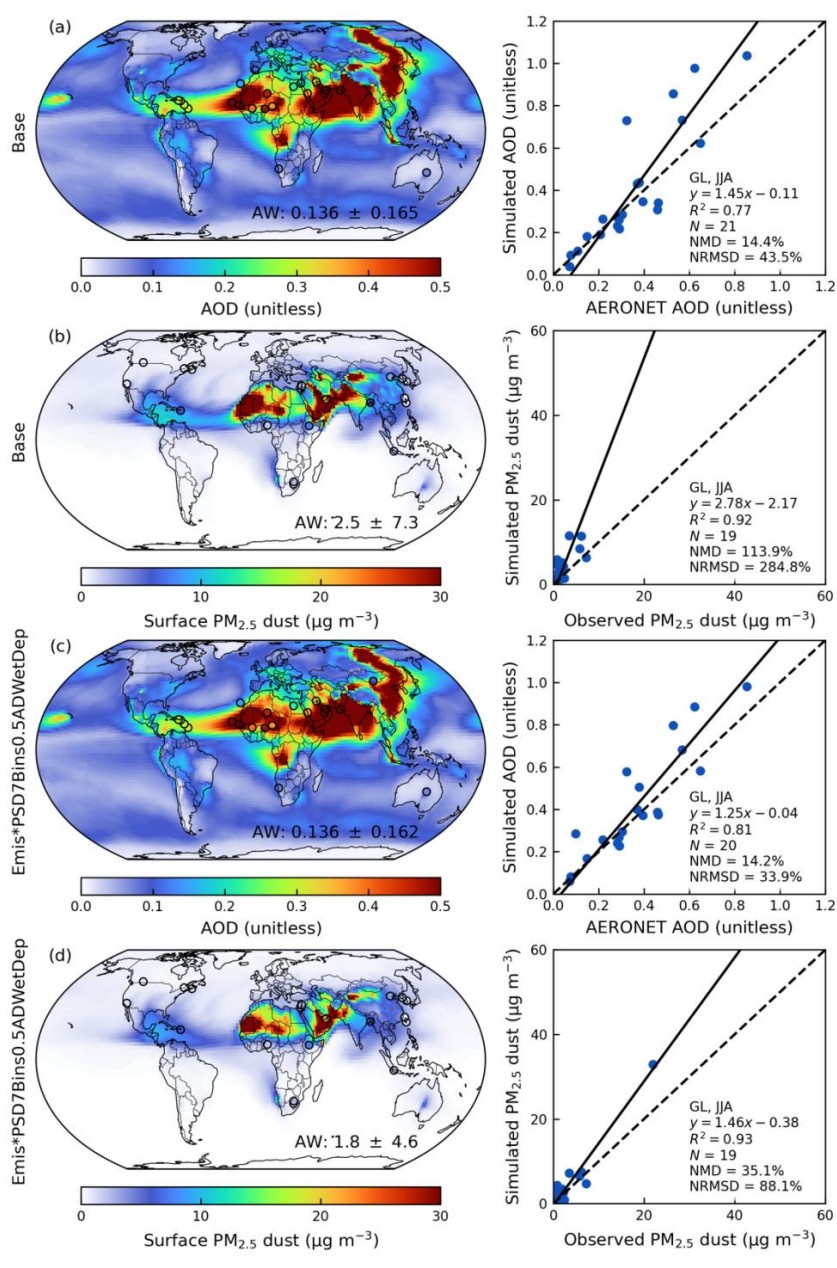


Figure A10. Same as Figure A9 but for the seasonal mean of June, July, and August (JJA).



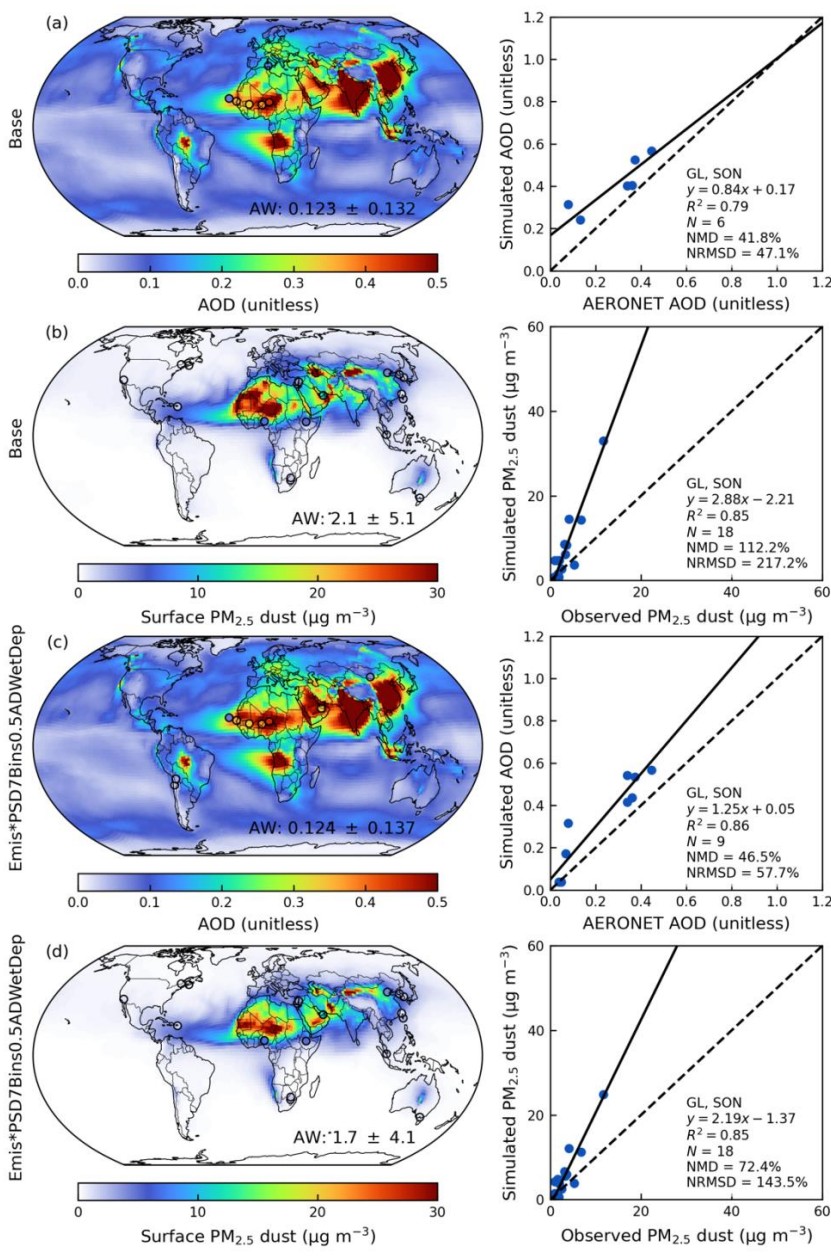


Figure A11. Same as Figure A9 but for the seasonal mean of September, October, and November
(SON).



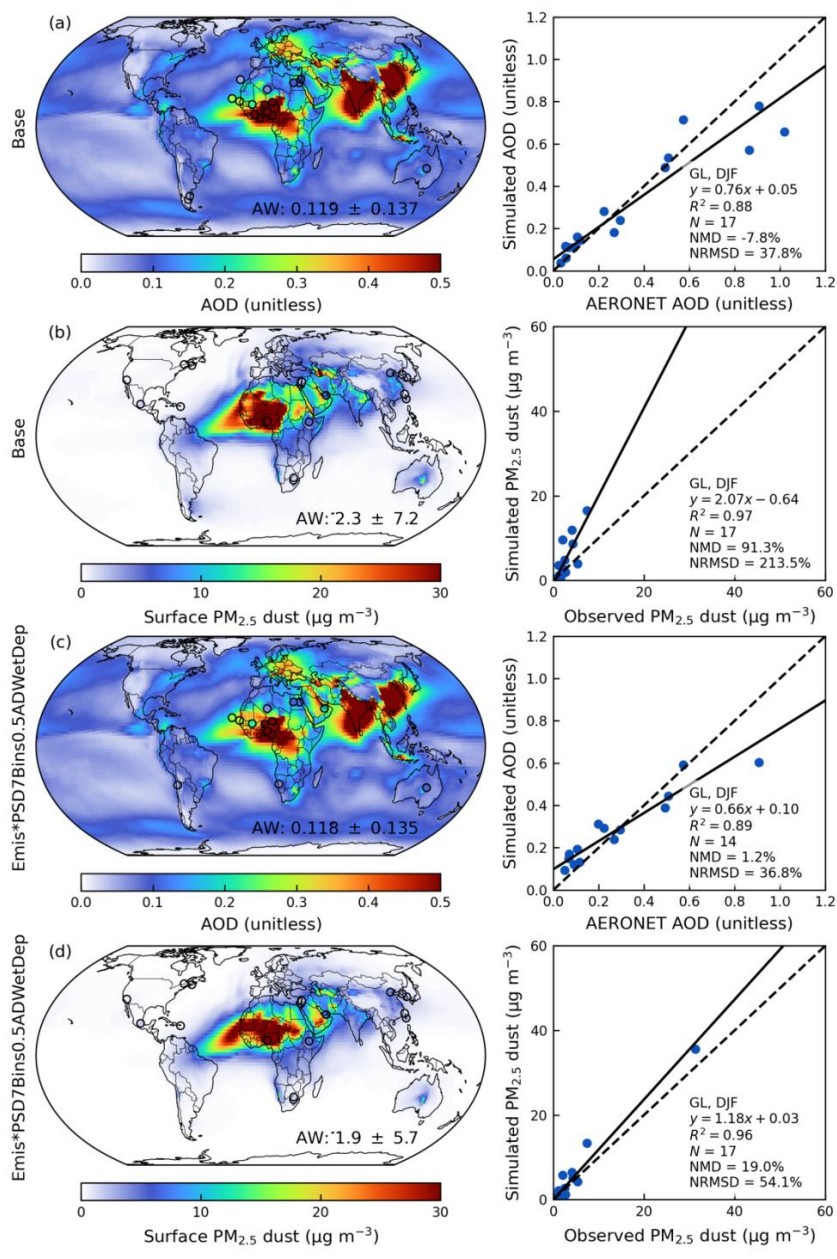


Figure A12. Same as Figure A9 but for the seasonal mean of December, January, and February (DJF).



839 ***Code availability***. The standard GEOS-Chem in its high-performance configuration version 14.4.1

840 can be downloaded at https://doi.org/10.5281/zenodo.12584305 (The International GEOS-Chem

841 User Community, 2024). The model source code, an example run directory, and the calculation

842 scripts for the hourly dust emission fluxes for the revised simulation can be downloaded at

843 https://doi.org/10.5281/zenodo.14510793 (Zhang, 2024).

844 ***Data availability***. The surface $PM_{2.5}$ dust measurements with the attenuation correction from

845 SPARTAN used in this study will be public available in future release at https://www.spartan-

846 network.org/data (last access: 4 February 2025). The $PM_{10}$ dust and total deposition of dust are

847 available at https://doi.org/10.5281/zenodo.6989502 (Li et al., 2022a). The processed

848 meteorological fields from GEOS-FP are available at

849 http://geoschemdata.wustl.edu/ExtData/GEOS_0.25x0.3125/GEOS_FP/ (last access: 4 February

850 2025) with the soil porosity downloaded from the constant land-surface parameter of MERRA2

851 M2C0NXLND collection (https://disc.gsfc.nasa.gov/datasets?project=MERRA-2, last access: 4

852 February 2025). The land cover dataset can be downloaded at

853 https://lpdaac.usgs.gov/products/mcd12c1v061/ (last access: 4 February 2025). The monthly

854 mean leaf area index at 0.5 degree can be downloaded at

855 http://globalchange.bnu.edu.cn/research/laiv6 (last access: 4 February 2025). The satellite-

856 derived aeolian roughness data are available upon contacting Catherine Prigent. The GSDE soil

857 dataset can be downloaded at http://globalchange.bnu.edu.cn/research/soilw (last access: 4

858 February 2025).

859 ***Author contribution***. The manuscript was written by DZ and RVM with contributions from all

860 authors. DZ and RVM designed the study with developments of the methodology. DZ conducted

861 simulations and analyzed the results. XL developed the methodology for the mineral dust

862 concentration construction in SPARTAN. AvD compiled the Deep Blue AOD dataset and ground-

863 based observation datasets of surface $PM_{2.5}$ dust over NA and AERONET AOD for evaluation. XL,

864 CRO and EW contributed to SPARTAN measurements. YL contributed to the dry deposition

865 analysis. JM offered valuable discussion for the emission scheme refinements. DML and JFK

866 contributed to the development of a new dust emission scheme. LL constructed the observational

867 data for $PM_{10}$ dust and deposition flux. HZ contributed to the generation of SPARTAN dust data. JRT

868 and YY contributed to the discussion of the evaluation of simulated dust. MB and YR contributed to

869 the establishment and maintenance of SPARTAN monitoring sites. All authors contributed to



revising the manuscript.
***Competing interests***. The authors declare no competing financial interest.
***Acknowledgements***. This work was supported by the National Science Foundation grants
2244984 and 2151093, and the National Aeronautics and Space Administration grant
80NSSC22K0200. The GEOS-FP data used in this study have been provided by the Global Modeling
and Assimilation Office (GMAO) at the NASA Goddard Space Flight Center. We thank the AERONET,
CALIOP, MODIS, and VIIRS teams for the creation and public release of their data products.

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
