# Peer review of "Improving Annual Fine Mineral Dust Representation from the Surface to the"

_EGUsphere, 2025_

## Author Response (AR1)

**Reviewer: 1**

*The manuscript titled GMD_2025_Improving Fine Mineral Dust Representation from the Surface to the Column in GEOS-Chem 14.4.1 describes the implementation of the latest dust emission scheme in the GEOS-Chem High Performance configuration (version 14.4). The authors emphasize the need to reconcile the modeled geometric diameter with in-situ measurements that are based on aerodynamic diameter. They also provide a comprehensive, step-by-step approach to refining and evaluating the model, effectively isolating uncertainties by region. The manuscript is well-organized, and the thoroughness of the performance evaluations is particularly commendable. Overall, I am impressed by the clarity and depth of this work, and I recommend acceptance with minor revisions.*

   Reply: We thank the reviewer for the affirmation of this work.

***Specific comments:***

1. ***Dust Optical Properties in GEOS-Chem****: It would be helpful to include a concise summary of the dust optical properties used in your GEOS-Chem configuration. You mention that the model employs the improved dust optical properties from Singh et al. (2024). Could you clarify whether the aspect ratios of the various dust bins in your model are consistent with those in Singh et al. (2024).*

   Reply: We adopted the dust optical properties, specifically the extinction coefficients and effective radii that are generated from Singh et al. (2024) for different dust size bins. We modified the description in line 192-196:

   "…updated optical properties for aspherical hydrophobic mineral dust (http://geoschemdata.wustl.edu/ExtData/CHEM_INPUTS/CLOUD_J/v2025-01/FJX_scat-aer.dat, last access: 7 April 2025) for different dust size bins as calculated by Singh et al. (2024) using the *T*-matrix method for an equiprobable mixture of prolate and oblate spheroids with varying aspect ratios using complex refractive indices from Sinyuk et al. (2003)."

2. ***Size Distribution Comparisons:*** *Your analysis would be more compelling if you compared the model's size distribution against in-situ measurements, including AERONET data and other publicly available datasets (e.g., doi.org/10.5194/essd-16-4995-2024). Such comparisons would provide additional evidence that your updates to the dust emission scheme accurately capture real-world size distributions.*

   Reply: We refer to the comparisons of the size distribution parametrization against various observations as shown in Figure 2 in Kok et al. (2011). On top of these prior comparisons, we compared with the measured size distribution of emitted dust from the Fennec field campaign in the North Africa that was recently available and covered the whole size range from 0.2 µm to 12 µm as shown in Figure 8a in our work. Comparison with AERONET size distributions would be a large complex undertaking as there is need to account for assumptions within the inversion process and for the size distribution of non-dust species.

 *Technical corrections:*

*In your figures, the circles appear to lack any visible color fill. Consider adjusting the visualization to ensure the colors are clear and distinguishable.*

Reply: We use the same color bar for the AOD or surface concentrations from simulations and ground-based measurements. The negligible contrast between them indicates that they are highly consistent with each other.

*Reporting "AW 2.3 ± 5.7" can be misleading because concentrations and emissions generally cannot be negative. Replacing the AW values with confidence intervals or alternative statistical measures might be more informative and intuitive for readers.*

Reply: We update all figures with the statistics of arithmetical mean with 5th and 95th percentiles in the square brackets to avoid confusion.

**Reviewer: 2**

*"Improving Fine Mineral Dust Representation from the Surface to the Column in GEOS-Chem 14.4.1" by Zhang et al. is a model development paper that evaluates the dust aerosol cycle in various configurations of the GEOS-Chem Chemical Transport Model. The paper is well-organized, although some key information is occasionally missing, which makes the paper difficult to fully evaluate. Most of my comments are requests for clarification. Overall, I found the paper illuminating and worth reading, especially as a dust scientist working with a different ESM. I am recommending that the paper be accepted subject to major revision. The authors can contact me at ron.l.miller@nasa.gov if they have any questions about my review.*

Reply: We thank Ron L. Miller for the constructive feedback, which helps strengthen this work.

*1. Some aspects of the model are incompletely documented. For example, the authors note that emission is calculated using winds from the high-resolution GEOS-FP Forward Processing model used for weather forecasting, whose horizontal resolution (0.25° latitude by 0.3125° longitude) is high compared to typical global dust models as well as the resolution of the GEOS-Chem transport model used here (closer to 2 degrees). However, I could not find how often emission is calculated: eight-times daily? Also, given that many studies nudge toward the GEOS MERRA2 product, could the authors tell us a little about the relation of the winds of the Forward Processing model and MERRA-2?*

Reply: We now add the frequency, specifically hourly, of the offline dust emissions in line 455-456:

"The calculated offline hourly dust emissions at $0.25^\circ \times 0.3125^\circ$ resolution using Equation (8) are then used to drive GCHP simulations at C48 resolution."

The main difference of GEOS-FP versus MERRA-2 for our simulation period of 2018 is finer spatial resolution to which dust emissions are highly sensitive. We update the statement for GEOS-FP meteorology in line 173-180 to emphasize the importance of the finer spatial resolution:

"The model is driven by meteorological inputs from GEOS Forward Processing (GEOS-FP; https://gmao.gsfc.nasa.gov/, last access: 4 February 2025) with a fine resolution $0.25^\circ \times 0.3125^\circ$ (~25 km) and 72 hybrid sigma-pressure vertical levels up to 0.01 hPa. GEOS-FP uses dynamic near-real-time assimilation algorithms compared to consistent static assimilation algorithms used in Modern-Era Retrospective analysis for Research and Applications Version 2 (MERRA-2; https://gmao.gsfc.nasa.gov/GMAO_products/, last access: 19 April 2025). We choose GEOS-FP over MERRA-2 for this study since GEOS-FP offers finer resolution for dust emission calculations."

*Similarly, the optical properties of dust particles are not fully described. The study by Singh et al. (2024) is cited ('updated optical properties for aspherical mineral dust'). I think this means that they are modeling dust as spheroids for scattering calculations, but they should clarify this as well as cite their source for the dust index of refraction. Singh et al., 2024 cite Tegen and Lacis, who use an index from measurements by Patterson et al. (1977). However, later Singh et al. cite the index compiled by Sinyuk et al. (2003). Please clarify how the index of refraction and optical properties were prescribed.*

*Patterson, E. M., D. A. Gillette, and B. H. Stockton (1977), Complex index of refraction between 300 and 700 nm for Saharan aerosols, J. Geophys. Res., 82, 3153–3160.*

*Sinyuk, A., O. Torres, and O. Dubovik, Combined use of satellite and surface observations to infer the imaginary part of refractive index of Saharan dust, Geophys. Res. Lett., 30(2), 1081, doi:10.1029/ 2002GL016189, 2003.*

Reply: Indeed, the dust optical properties are based on a mixture of spheroids and use refractive indices from Sinyuk et al. (2003). We update the statement for the specific optical properties we used and make the corresponding optical table publicly available in line 192-196:

"…updated optical properties for aspherical hydrophobic mineral dust (http://geoschemdata.wustl.edu/ExtData/CHEM_INPUTS/CLOUD_J/v2025-01/FJX_scat-aer.dat, last access: 7 April 2025) for different dust size bins as calculated by Singh et al. (2024) using the *T*-matrix method for an equiprobable mixture of prolate and oblate spheroids with varying aspect ratios using complex refractive indices from Sinyuk et al. (2003)."

*Removal processes could also be described in greater detail. Figure 9 shows the dry deposition velocity as a continuous function of particle size, even though it is prescribed in the model afor discrete bins. The same is true for the new washout parameterization. Figure 9 would be more useful if the authors plotted the discrete values of both the deposition speed and washout rates for each bin for both the 4 and 7-bin versions of the model. (i.e. replace the continuously varying washout rate with 7 discrete values for comparison to the two values used in the default model that are currently identified by an orange dashed line.)*

Reply: We update Figure 9 to include the discrete treatment implemented for 7-bin simulations.

*2. The authors adjust the global emission so that spatial variations in model AOD match those from the annual average Deep Blue AOD retrievals, according to a regression criterion (line 336). First, I cannot tell whether the base model is also calibrated in the same way.*

Reply: The base model is calibrated to have total annual dust emissions of ~2000 Tg yr$^{-1}$ with a global scaling factor and regional tuning factor in North America, which also shows a regression slope close to 1 (specifically 0.92) as indicated in Figure 6. We add a statement in line 219-223:

"The total dust emission flux in kg m$^{-2}$ s$^{-1}$ is calculated based on Zender et al. (2003) and Fairlie et al. (2007):

$$F_d = C_g C_{NA} f_{bare} S \varphi Q_s \qquad (1)$$

where $C_g$ is a global scaling factor and $C_{NA}$ is a regional scaling factor in North America for total annual emissions of ~2000 Tg yr$^{-1}$ as optimized by Meng et al. (2021)…"

*Second, a limitation of this method is that the model AOD depends not only upon dust but all the other aerosol types computed by GEOS-Chem. Thus, tuning the model to match Deep Blue may compensate for errors in these other fields and introduce biases into the dust AOD and the surface concentration. The model improvements described by the authors may not be entirely addressing limitations in the dust model, but rather biases in the other constituents. It is probably beyond the scope of this paper to address non-dust biases in AOD, but this uncertainty should be given greater emphasis in the method description and conclusions.*

Reply: We acknowledge that the AOD will be affected by other aerosol components and thus adopted a criterion of the fractional contribution of dust optical depth larger than 50% for the total aerosol optical depth. We add to the method section in line 134-136:

"We compare simulated AOD over mainly dusty regions (defined as $\text{AOD}_{\text{Dust}}/\text{AOD} > 0.5$ from simulations) against satellite and AERONET AOD to reduce the effects of errors in other AOD components and focus on the performance of mineral dust."

and the conclusions section in line 677-680:

"Despite these advances, challenges remain in mineral dust development and evaluation. The performance of AOD against satellite and AERONET observations over dusty regions may still be affected by other aerosol components which may benefit from further evaluations and developments."

*3. Another uncertainty that is not addressed is the temporal mismatch of the simulation period and measurements for evaluation. All the simulations are for the year 2018. In contrast, the CALIOP retrievals are for 2007-2021 while the SPARTAN dust PM2.5 network spans the five years between 2019 and 2023. (Some SPARTAN stations are based on as few as 10 measurements, which means that their annual average is subject to a potentially large sampling uncertainty.) This mismatch is partly the result of data availability, but it should at least be acknowledged in the conclusions as part of a fuller description of uncertainty.*

Reply: We attempt to address this source of uncertainty by averaging the data from CALIOP and SPARTAN for long-term representativeness. We add in line 680-685:

"Although the simulations are only for a single year, we average the multi-year observational data from the CALIOP extinction profile and SPARTAN measured surface dust concentrations for long-term representativeness. This approach benefits from the weak interannual variability of annual mean mineral dust concentrations (Li et al., 2017; Song et al., 2021). Nonetheless, additional observational data will enable further evaluation of the performance of mineral dust simulations."

*4. I like the careful comparison by the authors of the SPARTAN surface concentration measurements that are characterized by aerodynamic diameter to the geometric diameter used in the model. They note that even larger surface biases would result without this correction.*

Reply: We thank the reviewer for affirming the value of this work.

*This raises a question about the washout parameterization: (line 507, eq. 9). Is the dependence of washout rate to dust particle size derived or fitted assuming spherical particles? The authors are careful to use more irregular shapes when calculating optical properties (spheroids) and comparing to concentration measurements (ellipsoids?). Have the authors accounted for non-spherical shapes in their washout calculation? Ellipsoids and spheroids might have a greater chance of washout, comparable to spheres with larger diameter.*

Reply: We do not consider nonsphericity for washout since Ginoux et al. (2001) found that the effect is negligible for particles smaller than 6 μm which are the focus of our study. We add more clarity about our use of spherical particles in line 557-559:

"Using the same semi-empirical equations for *A* and *B* as Wang et al. (2014b), the updated values for different dust size bins with different effective spherical radii are summarized in Table 2."

*5. There are a few ad hoc assumptions that should be given more emphasis. The authors '[apply] a regional scaling factor of 0.6 over the Sahara to reduce its emissions' (line 367). This reduction seems arbitrary, and it has a big impact upon model behavior given the global importance of the North African source. Is there a physical basis for this? This rescaling should be acknowledged in the conclusions.*

Reply: We clarify the statement for regional scaling factor in line 449-452:

"Additionally, a regional scaling factor of 0.6 over the Sahara ($C_{sah}$) and unity elsewhere is applied to reduce regionally excessive dust emissions that may be influenced by the tendency for global models to overrepresent emissions from large source regions compared with smaller sources (Kok et al., 2021a; Zhao et al., 2022)."

*Similarly, the dependence of the wind speed threshold for emission upon soil moisture is treated somewhat arbitrarily. In particular, the authors reduce by half the soil moisture from the top layer of the GEOS-Chem land model before applying it to the reduction of the wind speed threshold for emission. In support, the authors cite Darmenova et al. (2009), along with Wu et al. (2022); the latter justify their own reduction by citing Darmeonova's doctoral thesis. Darmenova et al. point out that measurements linking the wind speed threshold and soil moisture are based upon the upper 1-2 cm. They note the coarser resolution of climate model soils, while citing measurements at different locations to argue that moisture near the surface should be less than moisture integrated over a deeper layer. The issue is that following precipitation, drying will start at the surface and propagate downward so that the upper 1-2 cm dry out first, before the deeper top layer in GEOS-Chem. Thus, as noted by Darmenova, the GEOS-Chem uppermost layer has too much inertia compared to the surface moisture that is observed to limit dust emission. The authors should present observations that show the relation of soil moisture at 1-2 cm compared to the 5-cm layer integral. Otherwise, they should acknowledge that their rescaling of the 5-cm GEOS-Chem value is a source of uncertainty.*

Reply: Thank you for this context. We rephrase line 441-445:

"Soil wetness is taken from the parent meteorological inputs of GEOS-FP (Koster et al., 2020) which targets the top 5 cm layer that desiccates more slowly following precipitation than the soil wetness in the top 1-2 cm layer (Swenson and Lawrence, 2014) that is most pertinent to dust emissions; we halve the soil wetness in an attempt to represent this process (Darmenova et al., 2009; Wu et al., 2022)."

We add acknowledgements for these uncertainties in line 685-688:

"In addition, knowledge gaps remain for mechanistic representation of mineral dust emissions. We call for further developments on the parametrization of dust emissions, particularly for the uncertainties in global and regional dust emission strength and further constraints on the effects of soil wetness on the threshold friction velocity."

*6. The Conclusions are brief and could be expanded to acknowledge uncertainties and the implications of results. For example, could you choose the dry deposition speed and the washout rate of DST1 in the 4-bin model so that it matches the removal rates of the 7-bin model? This would give you the improvement seen for 7 bins but with greater computational efficiency.*

Reply: We now add a new paragraph for the acknowledgments and uncertainties in line 677-688:

"Despite these advances, challenges remain in mineral dust development and evaluation. The performance of AOD against satellite and AERONET observations over dusty regions may still be affected

by other aerosol components which may benefit from further evaluations and developments. Although the simulations are only for a single year, we average the multi-year observational data from the CALIOP extinction profile and SPARTAN measured surface dust concentrations for long-term representativeness. This approach benefits from the weak interannual variability of annual mean mineral dust concentrations (Li et al., 2017; Song et al., 2021). Nonetheless, additional observational data will enable further evaluation of the performance of mineral dust simulations. In addition, knowledge gaps remain for mechanistic representation of mineral dust emissions. We call for further developments on the parametrization of dust emissions, particularly for the uncertainties in global and regional dust emission strength and further constraints on the effects of soil wetness on the threshold friction velocity."

Interesting idea about calibrating the 4-bin dust model. However, the computational cost of the 7-bin simulation is only 6% greater than that of the 4-bin simulation in GEOS-Chem. Thus, retaining the more mechanistic 7-bin approach seems advantageous, especially considering the continuing growth of computational resources. We add the computational cost for 4-bin and 7-bin dust simulations in Table A4.

*Also, the model is evaluated using annual average observations, but seasonal biases are relegated to the appendix. It would be helpful to have some discussion of seasonal model behavior including biases in the main article.*

Reply: We now expand the discussion for seasonal performance in line 603-605 while maintaining brevity to avoid distracting from our main analyses:

"The reduction of surface overestimation is especially prominent over dusty seasons in Spring (from 73% to 48%) and Summer (from 138% to 50%), while further improvements are needed for surface overestimation in Fall (from 140% to 95%)."

*Minor Comments:*

*73 'predicted spatial distribution \*of emission\*'*

Reply: We now update it in line 75-76:

"The predicted spatial distribution of dust emissions…"

*90 'an overestimation of fine dust (Kok, 2011; Kok et al., 2017)' see also Cakmur et al 2006*

*Cakmur, R.V., R.L. Miller, J.P. Perlwitz, I.V. Geogdzhayev, P. Ginoux, D. Koch, K.E. Kohfeld, I. Tegen, and C.S. Zender, 2006: Constraining the global dust emission and load by minimizing the difference between the model and observations. J. Geophys. Res., 111, D06207, doi:10.1029/2005JD005791.*

Reply: We add the citation as suggested in line 92:

"…an overestimation of fine dust (Cakmur et al., 2006; Kok, 2011; Kok et al., 2017)."

*96 'especially over size ranges with rapid variation in processes' Does 'rapid' refer to variations with respect to size?*

Reply: We update the sentence to avoid confusion in line 99:

"…especially with rapid variation in processes across different sizes."

*140 '10 samples for the 5-year' Have you tested a higher threshold for minimum measurement number? You are trying to resolve an annual average with only 10 samples over five years, which could be a problem if the measurements all occur within a particular season.*

Reply: We also test the comparison sensitivity of using SPARTAN data over sites with at least 50 samples, confirming the reduction of surface fine dust overestimation, which is summarized in Table A3.

The samples are generally distributed across seasons. We now update Table A1 to include observational seasons.

*158 'are computed with relative humidity dependent aerosol size distributions' Does the model calculate deliquescence of dust particles and its effect upon optical properties?*

Reply: Mineral dust is considered as hydrophobic. Relative humidity dependence is calculated for hydrophilic components. We update the sentence in line 188-196:

"The effects of aerosol on photolysis rates are computed with relative humidity dependent aerosol size distributions and optical properties for hydrophilic aerosols with improved parametrization for the effective radii of inorganic and organic aerosols (Latimer and Martin, 2019; Ridley et al., 2012; Zhu et al., 2023) and updated optical properties for aspherical hydrophobic mineral dust (http://geoschemdata.wustl.edu/ExtData/CHEM_INPUTS/CLOUD_J/v2025-01/FJX_scat-aer.dat, last access: 7 April 2025) for different dust size bins as calculated by Singh et al. (2024) using the *T*-matrix method for an equiprobable mixture of prolate and oblate spheroids with varying aspect ratios using complex refractive indices from Sinyuk et al. (2003)."

*166 'standard wet deposition scheme includes scavenging in convective updrafts, and in-cloud and below-cloud scavenging from precipitation.' What controls the rate of scavenging in convective updrafts? Is it different from the precipitation-rate dependence in stratiform clouds?*

Reply: Yes, there are separate parametrizations for convective and stratiform clouds. We rephrase line 200-202:

"Wet deposition includes separate algorithms for scavenging in convective updrafts, and in-cloud and below-cloud scavenging from precipitation (Liu et al., 2001; Wang et al., 2011, 2014a)."

*198 'where ƒ is the clay content in the top soil layer and a global constant value of 0.2 is used to reduce excessive sensitivity of dust emission fluxes to ƒ' This is unclear. Is fclay set equal to 0.2 or multiplied by 0.2?*

Reply: The clay content is set equal to 0.2. We now update the sentence in line 235:

"…the clay content in the topsoil layer and is set to a global constant value of 0.2…"

*203 'Brittle Fragmentation Theory (Kok, 2011) with parameter values optimized using dust observations from the Interagency Monitoring of Protected Visual Environments (IMPROVE) ground-based monitoring network in the United States' Is this correct? Zhang et al. (2013) say that the sidecrack propagation length of 8 um is taken from Zhao et al. (2010), who fitted to measurements of dust over North Africa during the DABEX field campaign.*

Reply: The purpose of tuning the size distribution was for better agreement against the observations with IMPROVE observations. We agree that the value of side crack propagation length is from the fitting of observations from Zhao et al. (2010) and update the sentence accordingly in line 239-242:

"The default size distribution of emitted dust in GEOS-Chem implemented by Zhang et al. (2013) is based on the Brittle Fragmentation Theory (Kok, 2011) with fitted parameter values for better agreement of dust observations from the Interagency Monitoring of Protected Visual Environments (IMPROVE) ground-based monitoring network in the United States…"

*237 'The mass fraction of each simulated dust size bin to the total fine dust mass concentrations can be calculated by the integration of the dust size distribution of Equation (4) with the $\lambda$ value of 12 µm of the default PSD used in the GEOS-Chem (GC PSD)' I'm confused, On the previous page (line 210), you quote 8 um as the default value. Second, eq. 4 describes the PSD of emission, but you are using it here to represent the PSD of load. Is this described correctly?*

Reply: This is unintentional. Corrected in line 276:

"…the $\lambda$ value of 8 µm of the default PSD…"

*280 'The simulated vertical profile shows excellent agreement against the 15-year (2007 to 2021)' To me, Figure 3 shows a consistent low bias of model dust in the lowest kilometer, with a corresponding high bias above? (This underestimate seems odd since model surface concentration is overestimated in comparison to SPARTAN: Figure 2).*

Reply: Agreed. We clarify in line 318-323:

"The simulated vertical profile exhibits overall agreement against the 15-year (2007 to 2021) climatological mean extinction vertical profile from the CALIOP, with no evidence of a model overestimate in the lower mixed layer versus aloft, indicating the vertical distribution of mineral dust is not the main driver of the performance discrepancy between the surface and the column. However, further evaluations of the vertical profile near the surface are needed as CALIOP retrievals are challenging at lower altitudes especially below 100 m."

*336 'The total global annual source strength for each sensitivity simulation is scaled to achieve unity slope versus Deep Blue AOD (Figure A1) over major dust source regions.' Is this also true for the BASE experiment? Is dust AOD (DAOD) calculated separately? What is the global DAOD for each experiment?*

Reply: As replied for comment #2, we add a statement for clarification of the dust emissions in the Base simulation in line 219-223:

"The total dust emission flux in kg m$^{-2}$ s$^{-1}$ is calculated based on Zender et al. (2003) and Fairlie et al. (2007):

$$F_d = C_g C_{NA} f_{bare} S \varphi Q_s \tag{1}$$

where $C_g$ is a global scaling factor and $C_{NA}$ is a regional scaling factor in North America for total annual emissions of ~2000 Tg yr$^{-1}$ as optimized by Meng et al. (2021)…"

We calculated dust optical depth with dust optical properties as stated in line 192-196:

"…updated optical properties for aspherical hydrophobic mineral dust (http://geoschemdata.wustl.edu/ExtData/CHEM_INPUTS/CLOUD_J/v2025-01/FJX_scat-aer.dat, last access: 7 April 2025) for different dust size bins as calculated by Singh et al. (2024) using the *T*-matrix method for an equiprobable mixture of prolate and oblate spheroids with varying aspect ratios using complex refractive indices from Sinyuk et al. (2003)."

We add global dust optical depth in Table 3.

*342 'Regression equations are calculated using reduced-major-axis linear regression' Could you give a brief description of reduced-major-axis linear regression and how it compares to the standard technique? How are you calculating the uncertainties for both the model and observations?*

Reply: We now add brief statement for the reason of choosing reduced-major-axis linear regression and a reference for interested readers to learn more in line 383-384:

"Regression equations are calculated using reduced-major-axis linear regression (Smith, 2009) to account for uncertainties in both simulations and measurements."

We now update uncertainty statistics to be more straightforward with the 5$^{th}$ and 95$^{th}$ percentiles.

*398 'In addition, we reduce the sensitivity of dust emissions to clay content by eliminating the multiplication of the capped clay content f' Wouldn't the removal of a factor less than one increase the sensitivity to clay? On line 198, you write that the addition of this factor is intended to reduce sensitivity.*

Reply: We remove the multiplication of the clay content to reduce the effects from it. We now replace the word "sensitivity" as "effects" to avoid confusion in line 439-440:

"In addition, we reduce the effects of clay content on dust emissions…"

Line 467-468:

"Eliminating the multiplication of the capped clay content of $f'_{clay}$ reduces the effects of the clay content…"

*400 'Soil wetness is taken from the parent meteorological inputs of GEOS-FP.' How reliable is the GEOS-FP soil wetness?*

Reply: We now add a reference for interested readers in line 441:

"Soil wetness is taken from the parent meteorological inputs of GEOS-FP (Koster et al., 2020)…"

*422 'Eliminating the multiplication of the capped clay content of f' reduces the dust emission sensitivity to the clay clay content, increasing emissions' see comment on line 398.*

Reply: As replied for the comment on line 398.

*435 'with improvements to the relative regional magnitude of dust across the Sahara, Middle East and Asia.' Is this true? According to the R2, NMD, NRMSD metrics in Figure 6, all the sensitivity experiments perform worse over Asia compared to the BASE simulation.*

Reply: The improvements of the relative regional magnitude are referring to more comparable regression slope among the Sahara, Middle East and Asia. We update the sentence in line 480-482:

"…with improvements to the relative regional magnitude of dust across the Sahara, Middle East and Asia as indicated by more comparable regression slopes (Figure 6)."

*Figure 6: why for each region does the number of observations vary according to the experiment? Is this because the number of locations where AODdust/AOD exceeds 0.5 varies across the sensitivity experiment? If so, perhaps note this in the caption?*

Reply: The variable number of observations is because of the criterion used. We discussed the effects of it in Figures A3 and A4 and in line 476-479:

"Using the same dusty region of the Base (Figure A3) or EmisClayWetLAI$_{thr}$ (Figure A4) scheme for the comparisons of all dust emission schemes versus Deep Blue AOD confirms similarly slight improvements of regional dust emissions."

We now modify the caption for Figure 6 to note it in line 500:

"Note the total number of points varies across different schemes."

*480 'we adopt the Kok PSD' This is confusing. You already adopt the Kok PSD in your BASE experiment. What you are changing here is the sidecrack propagation length estimated by Kok. I suggest identifying this experiment by the lambda value rather than 'Kok'.*

Reply: Rephrased to include $\lambda$ in line 527:

"…we adopt the Kok PSD with $\lambda$ of 12 μm…"

*504 'varying by 3 orders of magnitude for 505 diameter ranging from 1 to 10 μm' Wouldn't the contrast in the DST1 bin between 0.1 and 1 um be more relevant to fine dust?*

Reply: Rephrased to be more general in line 550-551:

"However, washout scavenging coefficients strongly depend on aerosol size (Wang et al., 2014b)."

*540 Please identify the 'ground-observations'.*

Reply: We now add the details about fine mineral dust observations in North America in line 160-168:

"Ground-based observations of PM$_{2.5}$ dust over North America are constructed with a global dust equation (Equation (A1); Liu et al., 2022) and the elemental measurements from the Air Quality System (AQS) database for speciated PM$_{2.5}$ observations in the United States (https://aqs.epa.gov/aqsweb/airdata/download_files.html#Daily, last access: 8 April 2025) and from the National Air Pollution Surveillance Program in Canada (https://data-donnees.az.ec.gc.ca/data/air/monitor/national-air-pollution-surveillance-naps-program/Data-Donnees/2018/?lang=en, last access: 8 April 2025). The AQS database includes measurements from both the Interagency Monitoring of Protected Visual Environments (IMPROVE) and Chemical Speciation Network (CSN) networks."

*549 'The simulated total column AOD would be underestimated by 14% compared to AERONET...' Is emission of the low-res model rescaled so that regression of model AOD versus Deep Blue has unity slope? More generally, the high performance version has been subjected to numerous adjustments to improve agreement with measurements and retrievals, so the low-res version is at a disadvantage.*

Reply: Yes, the scaled emissions are the same emissions that are used in the coarse-resolution (C48) simulation, so the concern raised is not applicable. We clarify at line 409-410:

"…scaling the global total emission flux to achieve unity regression slope of simulated AOD versus Deep Blue AOD over dusty regions."

This method exploits the resolution independence of emissions noted on lines 205-209:

"Offline emissions of lightning $NO_x$ (Murray et al., 2012), biogenic VOCs, soil $NO_x$, sea salt (Weng et al., 2020) and mineral dust (Sections 2.3 and 4.2) at $0.25° \times 0.3125°$ resolution are included to represent emission processes at the finest available resolution and to enable consistent emission fluxes across model resolutions."

*610 'overall consistency in the vertical shape' see comment on line 280*

Reply: As replied for the comment on line 280.

*619 'the Kok particle size distribution (PSD; Kok, 2011) better represents the mass fraction of fine dust measured during the Fennec field campaign over Northern Africa than the default PSD'. I agree that restoring the Kok value of lambda to 12 um gives better results in GEOS-Chem, but I think the FENNEC results are being oversold as a justification for the change. The difference of the Kok and GC PSD in Figure 8 are small compared to the large spread of the FENNEC emitted size distribution.*

Reply: We agree that there are large uncertainties from observations of the size distribution of emitted dust as well. We update the statement to note the uncertainty from observations in line 670-672:

"…better represents the mass fraction of fine dust measured during the Fennec field campaign over Northern Africa than does the default PSD despite the uncertainties from the Fennec observations."

**Reviewer: 3**

*The paper titled "Improving Fine Mineral Dust Representation from the Surface to the Column in GEOS-Chem 14.4.1" aims to enhance the simulation of fine dust in the GEOS-Chem model using the GCHP framework by updating dust-related parameters and implementing a new dust scheme. There are some interesting results, However, as an evaluation of global dust model performance, some important details are not clearly described, and I have significant concerns regarding the methodology, particularly the limited number of observational sites used for validation and the reliance on single annual mean values. With only ~16–26 globally distributed sites (from AERONET and SPARTAN) providing one annual value each for 2018, the statistical metrics (e.g., R², NMD, NRMSD) lack robustness and may not reliably represent global model performance. For meaningful dust model evaluation—especially in the context of dust prediction—it is more appropriate to assess the model's ability to capture daily variability over key dust source regions rather than depending solely on globally averaged annual mean comparisons at sparse locations. I encourage the authors to address this limitation explicitly and consider incorporating higher temporal resolution evaluations and more regionally focused diagnostics to strengthen the assessment. I therefore recommend a **major revision** and suggest that the manuscript be resubmitted after the following comments and questions have been thoroughly addressed.*

Reply: We thank the reviewer for sharing this perspective. However, our objective is to address global model bias, rather than the different objective of "ability to capture daily variability over key source regions". We clarify our objective by revising the title:

"Improving Annual Fine Mineral Dust Representation from the Surface to the Column in GEOS-Chem 14.4.1"

abstract:

line 28:

"…annual mean surface dust concentrations…"

line 29:

"…improving the annual simulation…"

line 34:

"…on annual mean concentrations."

introduction:

line 54:

"…accurate representation of long-term concentrations…"

line 62:

"…the annual mean…"

adding a new paragraph in the introduction line 106-108 to clarify our objective:

"Many studies have examined daily dust variability for the purpose of short-term prediction (Amato et al., 2013; Tindan et al., 2023; Yu et al., 2021). Our study focuses on a different objective of accuracy of annual mean concentrations."

line 116-117:

"We focus on improving the annual fine dust representation in GCHP…"

also modifying conclusions line 652:

"…improve the annual mineral dust simulation…"

We also add to Table A1 the number of days in which a SPARTAN sample was taken. All sites had samples taken on at least 100 days, for a total of 10,072 days with samples, thus affirming the robustness of the measurements and statistics. AERONET also has a large number of measurements at each site. We add at line 123-124:

"The median number of days with AERONET measurements is 168 days for each site."

1. *L133: It is unclear whether the SPARTAN data used in this study represent daily means, monthly means, or another temporal interval. Additionally, the temporal resolution of the AERONET data should be clearly described to ensure consistency and clarity in the model–observation comparison.*

   Reply: We add the sampling and averaging frequency of SPARTAN data in line 145-159:

   "The sampling station follows either a standard sampling protocol or the National Aeronautics and Space Administration (NASA) – Italian Space Agency (ASI) Multi-Angle Imager for Aerosols (MAIA) sampling protocol. Under the standard sampling protocol, PM$_{2.5}$ is collected at staggered 3-hour intervals over a 9-day period, generating a 24-hour PM$_{2.5}$ sample covering a full diel cycle. Under the MAIA sampling protocol, PM$_{2.5}$ is collected continuously for 24 hours from 9 am to 9 am at a mission-defined frequency, which has been typically every 3 days during the sampling periods used here. The starting dates for MAIA sites are listed in Table A1. SPARTAN samples are analyzed for fine mineral dust concentrations using X-ray Fluorescence (XRF) and a global mineral dust equation (Equation (A1); Liu et al., 2022) including correction of attenuation effects due to mass loading. The 5-year averaged surface fine dust concentrations from SPARTAN sites are listed in Table A1. We use data from sites with at least 10 samples for the 5-year (2019–2023) period after the network began using XRF with samples. A sensitivity analysis requiring at least 50 samples per site is also conducted. This study used 2,296 filters from 25 SPARTAN sites for a total of 10,072 observational days."

   We add the AERONET data frequency in line 123-125:

   "The median number of days with AERONET measurements is 168 days for each site. We average daily AERONET AOD to an annual mean in the year of 2018."

2. *L239: In the default PDF used in the GEOS-Chem (GC PDS), the lambda is not 12, however 8, which is implemented by Zhang et al., 2013. Please double check the correct values and calculation.*

   Reply: Corrected in line 275-276:

   "…dust size distribution of Equation (4) with the $\lambda$ value of 8 μm of the default PSD used in the GEOS-Chem (GC PSD)…"

3. *L257 and Fig. 1: In GEOS-Chem and other models, PM2.5 dust is typically estimated using an empirical formula (e.g., bin1 + 0.38 × bin2) based on assumed size distributions within each bin. This approach differs from observational definitions of PM2.5, which are determined by instrument-specific inlet penetration efficiencies (e.g., WINS or SCC). In Figure 1, the authors appear to apply such instrument-based cut-off functions to define PM2.5 dust fractions (e.g., stating 67.6% of DST1 based on SCC 1.829), which differs from the standard model approach. However, it is unclear whether this correction is actually implemented in the model calculations or only used for comparison with observations. I suggest the authors clarify whether and how this method was applied, and if so, describe it explicitly in the methods section.*

Reply: We state our calculation of $PM_{2.5}$ dust in the Base simulation in line 291-293:

"In our Base simulation using the standard version of GEOS-Chem, we calculate surface $PM_{2.5}$ dust as 67.6% of DST1 to account for both aerodynamic diameter and inlet collection efficiency."

4. *Section 3: Again, the method used by the authors to calculate PM2.5 dust could significantly affect the subsequent comparison with observations. It remains unclear whether the apparent overestimation of PM2.5 dust is due to differences in how PM2.5 dust concentration is defined or calculated in the model.*

Reply: As replied for comment #3. We state our calculation of $PM_{2.5}$ dust in the Base simulation in line 291-293:

"In our Base simulation using the standard version of GEOS-Chem, we calculate surface $PM_{2.5}$ dust as 67.6% of DST1 to account for both aerodynamic diameter and inlet collection efficiency."

Thus, differences are driven by how $PM_{2.5}$ dust concentration is calculated.

5. *L286: The (AODdust/AOD ) >0.5 is based on GEOS-Chem model prediction or based on AERONET observation? The AOD comparison is also 2018 annual?*

Reply: We clarify our caption for Figure 2 in line 325-327:

"Annual simulated aerosol optical depth (AOD) and comparison against ground-based observations from AERONET over dusty regions (simulated $AOD_{Dust}/AOD > 0.5$) (top) in the year of 2018;"

6. *Figure 2. In relation to the previous point, there is also uncertainty in the ratio of AOD_dust to total AOD between the model and observational data. Many previous studies have shown that significant discrepancies in species ratios between model outputs and observations can strongly affect the reliability of model evaluation. However, the authors do not discuss this issue, nor do they provide any supporting evidence or analysis related to it.*

Reply: We use the criterion of simulated $AOD_{Dust}$/AOD > 0.5 to focus on the model performance over dusty regions. We add clarification for this criterion in line 134-136:

"We compare simulated AOD over mainly dusty regions (defined as $AOD_{Dust}/AOD > 0.5$ from simulations) against satellite and AERONET AOD to reduce the effects of errors in other AOD components and focus on the performance of mineral dust."

We also add in the conclusions line 677-680:

"Despite these advances, challenges remain in mineral dust development and evaluation. The performance of AOD against satellite and AERONET observations over dusty regions may still be affected by other aerosol components which may benefit from further evaluations and developments."

7. *Line 288: According to line 140, the study uses SPARTAN data from sites with at least 10 samples over the 5-year period (2019–2023) since the network began using XR. This raises concerns about the data quality used for comparison and validation in Figure 2 and the subsequent analysis. In particular, it is unclear how many valid observations are available for the PM2.5 dust comparison shown in Figure 2. Beyond the scatter plots, I strongly recommend that the authors include time series comparisons at several representative sites near dust source regions. Scatter plots between annual model simulations and observations may be misleading, especially if each site has only a few observations throughout the entire year.*

Reply: We now clarify in the title, abstract, introduction, and conclusion that our objective is focused on annual mean dust concentrations. Our focus is motivated by the importance of long-term dust concentrations, and by the bias that we found in these long-term concentrations versus new SPARTAN data. We clarify our objective by revising the title:

"Improving Annual Fine Mineral Dust Representation from the Surface to the Column in GEOS-Chem 14.4.1"

abstract:

line 28:

"…annual mean surface dust concentrations…"

line 29:

"…improving the annual simulation…"

line 34:

"…on annual mean concentrations."

introduction:

line 54:

"…accurate representation of long-term concentrations…"

line 62:

"…the annual mean…"

adding a new paragraph in the introduction line 106-108 to clarify our objective:

"Many studies have examined daily dust variability for the purpose of short-term prediction (Amato et al., 2013; Tindan et al., 2023; Yu et al., 2021). Our study focuses on a different objective of accuracy of annual mean concentrations."

line 116-117:

"We focus on improving the annual fine dust representation in GCHP…"

also modifying conclusions line 652:

"…improve the annual mineral dust simulation…"

We average SPARTAN measurements over 5 years to ensure data quality, which is very important for the evaluation of model performance as stated by the reviewer. The 9-day SPARTAN sampling protocol is designed for long-term rather than daily assessment. We also add to Table A1 the number of days in which a SPARTAN sample was taken. All sites had samples taken on at least 100 days, for a total of 10,072 days with samples, thus affirming the robustness of the measurements and statistics. AERONET also has a large number of measurements at each site. We add at line 123-124:

"The median number of days with AERONET measurements is 168 days for each site."

The comparisons against globally distributed measurements of $PM_{2.5}$ dust from SPARTAN, globally distributed measurements of $PM_{10}$ dust (Figures A8 and A9), and regionally distributed measurements of $PM_{2.5}$ dust over North America (Figures A7) all show better or comparable performance for the new developments compared to the default treatment. We also add to the conclusions a recommendation to examine daily variations in future work in line 688-689:

"Future examination of daily variability would also be valuable for short-term predictability."

8.  *Figure2: Figure 2 provides useful regression statistics such as NMD and NRMSD for comparing simulated AOD and surface PM2.5 dust against ground-based observations, the number of sites used in the analysis (approximately 16 for AERONET and 26 for SPARTAN) is relatively limited. Since each site contributes only a single annual value, the statistical robustness and global representativeness of these metrics may be limited. I suggest the authors discuss this limitation explicitly and consider including additional evaluations, such as seasonal or monthly comparisons if available, or using regional breakdowns to enhance the interpretability of the model performance.*

    Reply: We also add to Table A1 the number of days in which a SPARTAN sample was taken. All sites had samples taken on at least 100 days, for a total of 10,072 days with samples, thus affirming the robustness of the measurements and statistics. AERONET also has a large number of measurements at each site. We add at line 123-124:

    "The median number of days with AERONET measurements is 168 days for each site."

    Annual means are necessary to address our objective of evaluating spatial rather than temporal variations. Nonetheless, analyses for seasonal comparisons can be found in Figures A11 to A14 and text in line 599-605:

    "Overall comparisons for the seasonal mean between the Base and the Emis*PSD7Bins0.5ADWetDep simulations confirm largely reduced overestimation for the surface fine dust against SPARTAN, while retaining comparable skill for the total column AOD against AERONET (Figures A11 to A14). The reduction of surface overestimation is especially prominent over dusty seasons in Spring (from 73% to 48%) and Summer (from 138% to 50%), while further improvements are needed for surface overestimation in Fall (from 140% to 95%)."

    Analyses for regional performance can be found in Figures 5 and 6 and associated discussion for them when comparing against Deep Blue satellite AOD. As stated by the reviewer, there are too few

measurement sites from AERONET and SPARTAN for regional discussions for these comparisons. We emphasize in the conclusions the need for more observational data in line 684-685:

"Nonetheless, additional observational data will enable further evaluation of the performance of mineral dust simulations."

9. *Figure 3: The use of extinction profiles normalized by AOD introduces a scaling effect that can obscure absolute errors in the extinction values. While such normalization is helpful for assessing the relative vertical structure of aerosols, it is less effective for diagnosing biases that contribute to discrepancies in column vs. surface aerosol performance. A direct comparison of extinction values would more clearly indicate whether vertical profile inaccuracies — in both shape and magnitude — are responsible for the model's surface vs. column performance differences. It would be helpful if the authors could clarify this choice and ideally provide a comparison using the unnormalized extinction as well.*

    Reply: We now add Figure A1 for the absolute extinction vertical profile. Our conclusions remain unchanged.

10. *L307: The authors concluded that the model overestimates PM2.5 dust based on a comparison with the 2018 annual mean from SPARTAN data using the scattering method. However, this conclusion is not comprehensive, as it does not incorporate other ground-based observations to assess the uncertainty of SPARTAN. Additionally, the authors do not provide information on data availability or temporal variability. I recommend validating this conclusion using additional PM2.5 dust datasets, such as AIRNOW or IMPROVE, to enhance the robustness of the assessment.*

    Reply: We did not find other publicly available ground-based measurements available in dusty regions over the Sahara, Middle East, and Asia. The availability of SPARTAN data is provided in Data Availability section. Comparisons against other dust observations such as IMPROVE are included for ground-based measurements of $PM_{2.5}$ dust over North America in Figure A7, and $PM_{10}$ dust globally in Figures A8 and A9. We clarify at line 166-168:

    "The AQS database includes measurements from both the Interagency Monitoring of Protected Visual Environments (IMPROVE) and Chemical Speciation Network (CSN) networks."

11. *L315 and Figure 4: Could you please clarify the size distribution used for the default dust simulation (base) in GEOS-Chem? It would be helpful to know the specific aerosol size bins or the distribution function applied in the model for dust particles.*

    Reply: The details about the size distribution parametrization for the Base simulation and corresponding size bins are included in the main text of section 2.4 line 239-248:

    "The default size distribution of emitted dust in GEOS-Chem implemented by Zhang et al. (2013) is based on the Brittle Fragmentation Theory (Kok, 2011) with fitted parameter values for better agreement of dust observations from the Interagency Monitoring of Protected Visual Environments (IMPROVE) ground-based monitoring network in the United States:

$$\frac{dV_d}{d \ln D_d} = \frac{D_d}{c_V}\left[1 + \mathrm{erf}\left(\frac{\ln(D_d/\overline{D_s})}{\sqrt{2}\ln \sigma_s}\right)\right]\exp\left[-\left(\frac{D_d}{\lambda}\right)^3\right] \tag{4}$$

    where $V_d$ is the normalized volume for emitted dust aerosols in diameter of $D_d$ in μm; $c_V$ is the normalization constant to make the integration total of $V_d$ of 1; $\overline{D_s} = 3.4$ μm is the median diameter of

soil particles; $\sigma_s = 3.0$ is the geometric standard deviation of soil particles; $\lambda$ is the side crack propagation length, whose value is 8 μm in the default particle size distribution (PSD) used in the GEOS-Chem (GC PSD), and is 12 μm in the Kok PSD (Kok, 2011). "

12. *L328: What is the value of Lamda, 12? Did you compare with the result of using 8 in the default GC dust scheme?*

Reply: Add in line 369:

"…length of $\lambda$ (12 μm versus 8 μm)…"

13. *Figure 6: In Figure 6, the number of validation grid points (N) differs across the dust experiments due to the use of a dust-region mask defined by AOD_dust/AOD > 0.5. I believe this approach is not entirely reasonable, as the dust source regions themselves should remain consistent when using the same dust source function. The variations in AOD across experiments are a result of changes in dust scheme parameters, not the geographic extent of dust source regions. By applying the AOD_dust/AOD > 0.5 threshold independently for each experiment, you may be artificially excluding regions where the AOD has been reduced due to your parameter changes. This not only alters the spatial domain being compared but also potentially removes part of the signal you aim to evaluate—namely, the impact of those parameter changes on AOD. As a result, the comparison across experiments is not entirely fair and may underestimate the full effect of the modified parameters on dust AOD. To ensure a fair comparison, I suggest to be based on the BASE run for the dust source region instead of Figure A2.*

Reply: We now add Figure A3 to use the same dust source regions for the Base scheme for all dust emission scheme comparisons. Our conclusions remain unchanged.

14. *Figure 7: I would like to raise the same concern as using this validation methor to get conclusion that the performance are improve as : the number of sites used in the analysis (approximately 16 for AERONET and 26 for SPARTAN) is relatively limited. Since each site contributes only a single annual value, the statistical robustness and global representativeness of these metrics may be limited. I suggest the authors discuss this limitation explicitly and consider including additional evaluations, such as seasonal or monthly comparisons if available, or using regional breakdowns to enhance the interpretability of the model performance.*

Reply: As replied for the general comment and comments #7 and #10, our analyses are subject to limited availability of dust observations. We average data to ensure high data quality and the fairness for the evaluation of model performance. AERONET data is averaged from daily observations to ensure high quality in line 124-125:

"We average daily AERONET AOD to an annual mean in the year of 2018."

The quality of SPARTAN data is emphasized in line 145-159:

"The sampling station follows either a standard sampling protocol or the National Aeronautics and Space Administration (NASA) – Italian Space Agency (ASI) Multi-Angle Imager for Aerosols (MAIA) sampling protocol. Under the standard sampling protocol, $PM_{2.5}$ is collected at staggered 3-hour intervals over a 9-day period, generating a 24-hour $PM_{2.5}$ sample covering a full diel cycle. Under the MAIA sampling protocol, $PM_{2.5}$ is collected continuously for 24 hours from 9 am to 9 am at a mission-defined frequency, which has been typically every 3 days during the sampling periods used here. The starting dates for MAIA sites are listed in Table A1. SPARTAN samples are analyzed for fine mineral dust

concentrations using X-ray Fluorescence (XRF) and a global mineral dust equation (Equation (A1); Liu et al., 2022) including correction of attenuation effects due to mass loading. The 5-year averaged surface fine dust concentrations from SPARTAN sites are listed in Table A1. We use data from sites with at least 10 samples for the 5-year (2019–2023) period after the network began using XRF with samples. A sensitivity analysis requiring at least 50 samples per site is also conducted. This study used 2,296 filters from 25 SPARTAN sites for a total of 10,072 observational days."

We also add to Table A1 the number of days in which a SPARTAN sample was taken. All sites had samples taken on at least 100 days, for a total of 10,072 days with samples, thus affirming the robustness of the measurements and statistics. AERONET also has a large number of measurements at each site. We add at line 123-124:

"The median number of days with AERONET measurements is 168 days for each site."

We also test the comparison sensitivity of using SPARTAN data over sites with at least 50 samples, confirming the reduction of surface fine dust overestimation, which is summarized in Table A3.

We additionally call for more observations over global dusty regions in the conclusion section of line 680-685:

"…we average the multi-year observational data from the CALIOP extinction profile and SPARTAN measured surface dust concentrations for long-term representativeness. This approach benefits from the weak interannual variability of annual mean mineral dust concentrations (Li et al., 2017; Song et al., 2021). Nonetheless, additional observational data will enable further evaluation of the performance of mineral dust simulations."

15. *L437-442: Given the relatively small number of ground-based sites used in the evaluation (with only one annual mean value per site), I would caution against drawing strong conclusions from the changes observed in Figure 7. While the reduction in mean bias in surface PM2.5 dust is notable, the accompanying decrease in R² for both AOD and PM2.5 suggests a potential loss in spatial agreement. This appears to contradict the claim that AOD skill "remains comparable" to the base simulation. I recommend that the authors either provide statistical support for this conclusion (e.g., uncertainty estimates or significance testing), or qualify the statement to reflect the limitations of the evaluation dataset.*

Reply: Figure 7 shows intermediate developments for the dust emission scheme. We refer to the final development as evaluated for the simulation of Emis*PSD7Bins0.5ADWetDep in Table 3. The correlation coefficient for AOD and PM$_{2.5}$ dust all shows comparable performance for AOD and improved performance in PM$_{2.5}$ dust. We call for more observations over global dusty regions in the conclusion section of line 680-685:

"…we average the multi-year observational data from the CALIOP extinction profile and SPARTAN measured surface dust concentrations for long-term representativeness. This approach benefits from the weak interannual variability of annual mean mineral dust concentrations (Li et al., 2017; Song et al., 2021). Nonetheless, additional observational data will enable further evaluation of the performance of mineral dust simulations."

16. *Figure 11: Although the reduced-major-axis linear regression slope for PM2.5 dust against SPARTAN observations is further reduced from 1.53 (Figure 8) to 1.44, and the NMD remains comparable, I am concerned that this apparent improvement in surface PM2.5 dust may come at the cost of degraded*

*performance in AOD. Specifically, the NMD and NRMSD values for AOD in this experiment appear to have increased significantly compared to those in Figure 2. However, the actual NMD and NRMSD values for AOD in Figure 8 are not explicitly reported, making it difficult to assess the trade-off between surface and column performance. I recommend that the authors clearly report the AOD NMD and NRMSD values in Figure 8 to support a more complete and transparent evaluation of model performance.*

Reply: We summarize all statistics for AOD and PM$_{2.5}$ dust evaluations in Table 3.

17. *Figure A5: I noticed that the base run shows the best agreement in terms of the regression slope for PM2.5 dust against SPARTAN observations, with a value closest to 1.0. This indicates strong consistency in magnitude between the model and observations, which is particularly important for evaluating exposure-relevant quantities such as surface PM2.5 dust. While other runs may show improvements in NMD or NRMSD, I would encourage the authors to clarify which performance metric they consider most important in this context, and why. A more balanced discussion of the trade-offs between slope, bias, and correlation would strengthen the interpretation of model performance.*

Reply: Figure A5 (now Figure A7) is for the evaluation over North America, which is a much weaker dust source region compared to the Sahara, Middle East, and Asia. Our focus is for the model performance of AOD and surface PM$_{2.5}$ dust over global dusty regions. Figure A7 should be treated as complementary with slight improvements in the correlation.

18. *Figure A6-7: I am unclear about the meaning of "Emis*" as referenced in the figure caption. Could the authors clarify what this experiment represents and how it differs from the base simulation?*

Reply: All sensitivity simulation setup was summarized in Figure 4 and text in section 4.1 and was named to be self-indicative intentionally. Line 354-376:

"Figure 4 summarizes the setup of sensitivity simulations to evaluate the effects of algorithmic modifications and their performance versus satellite-retrieved AOD and surface dust measurements. The default dust simulation (Base) in GEOS-Chem as implemented by Fairlie et al. (2007) uses the DEAD emission scheme (Zender et al., 2003) with a topographical source function (Ginoux et al., 2001; Meng et al., 2021) for natural dust (GC Dust) with 4 dust size bins for emission, transport and removal with 7 dust size bins for dust optical depth calculation and heterogeneous chemistry. To improve the spatial distributions of dust total column abundance, we implement a new dust emission scheme developed by Leung et al. (2023) (DustL23; Emis). Additional modifications on top of the original DustL23 emission scheme include 1) reducing the sensitivity of soil clay content by eliminating the multiplication of the factor of the capped soil clay content $f'_{clay}$ (EmisClay); 2) halving the topmost soil wetness in the layer of 0-5 cm to approximate the soil wetness in the top 1-2 cm layer which is most pertinent to dust emissions (Darmenova et al., 2009; Wu et al., 2022) (EmisClayWet); and 3) reducing the threshold of $LAI_{thr}$ from 1.0 m$^2$ m$^{-2}$ to 0.5 m$^2$ m$^{-2}$ (EmisClayWetLAI$_{thr}$ or Emis*). To further improve the surface fine dust simulation, we update the GEOS-Chem particle size distribution (PSD) with the PSD developed by Kok et al. (2011) (Emis*PSD) with a larger value for the side crack propagation length of $\lambda$ (12 μm versus 8 μm) which reduced the mass fraction of emitted fine dust. The Kok PSD was shown to have excellent agreement versus various soil size measurements (Kok, 2011), especially for fine dust distributions (González-Flórez et al., 2023). Lastly, we allow for the four dust bins with geometric diameter less than 2 μm to have separate emission, transport, and dry and wet deposition while halving anthropogenic dust emissions from AFCID (Emis*PSD7Bins0.5AD), and with updated below-cloud or washout scavenging parametrization (Emis*PSD7Bins0.5ADWetDep). Each of these changes is examined below."

19. *Table 3: While the Emis* and subsequent sensitivity experiments reduce the NMD and slope for SPARTAN PM2.5 dust, these changes are accompanied by increased NMD for AERONET AOD. This suggests a trade-off between surface and column performance, which should be explicitly acknowledged and discussed. Deep Blue AOD performance is unchanged: Across all simulations, the correlation (r), slope, and NMD versus Deep Blue AOD remain fixed (e.g., r = 1.00, slope = 1.00 for many runs), which is unexpected. Please priove a brief explanation of why these metrics remain unchanged. It would be helpful to indicate the number of observation sites (N) used in the calculation of these statistics for each metric, especially since previous figures suggested limited spatial coverage (e.g., <20 sites).*

Reply: We respectfully think the performance change for AOD is minor compared to significant reduction of surface $PM_{2.5}$ dust as shown in Table 3. The dust emissions are scaled to achieve unity regression slope against Deep Blue satellite AOD as stated already in line 409-410:

"…scaling the global total emission flux to achieve unity regression slope of simulated AOD versus Deep Blue AOD over dusty regions."

Details about SPARTAN observations, including the number of observations at each site, are summarized in Table A1.

20. *L539-544: This paragraph, along with the corresponding figures in the Appendix, is difficult to follow. It is unclear why the authors conducted the C48 runs, what specific purpose they serve, and which figures are based on C48 versus C49 configurations. The motivation, methodological details, and relevance of these results are not clearly explained, making it challenging for readers to understand their significance. I strongly recommend that the authors either substantially revise this section to clearly articulate the purpose, setup, and interpretation of the C48 experiments, or consider removing it altogether if it is not essential to the main conclusions.*

Reply: We did not discuss results with dust emissions at C48 resolution in line 539-544. Perhaps the reviewer is referring to line 547-551. We also did not conduct a C49 configuration. We add the test at C48 resolution for the spatial resolution effects on dust emissions, which is consistent with prior studies. We modify the motivation in line 594-599:

"Consistent with prior studies about the spatial sensitivity of dust emissions (Leung et al., 2023; Meng et al., 2021), fine-resolution meteorological fields are needed to capture dust emission hotspots. If the dust emissions were calculated with C48 meteorological fields, the global dust distribution would become more concentrated in the major global source regions with the elimination of marginal dust sources, and the $R^2$ versus SPARTAN surface $PM_{2.5}$ dust would diminish to 0.83 (Figure A10)."

21. *Figure A9-12:The validation of seasonal mean values is based on a very limited number of observational sites—fewer than 20, and in some cases fewer than 10. This small sample size raises concerns about the statistical robustness and representativeness of the calculated performance metrics (e.g., $R^2$, NMD, NRMSD). With such limited spatial coverage, the results may be highly sensitive to outliers or regional biases, and may not reliably reflect the model's performance at larger scales. I recommend that the authors either supplement the evaluation with additional datasets or explicitly discuss the limitations associated with the small sample size.*

Reply: As replied for the general comment and comments #7 and #10, our analyses are subject to limited availability of dust observations. We average data to ensure high data quality and the fairness for the evaluation of model performance. The quality of SPARTAN data is emphasized in line 145-159:

"The sampling station follows either a standard sampling protocol or the National Aeronautics and Space Administration (NASA) – Italian Space Agency (ASI) Multi-Angle Imager for Aerosols (MAIA) sampling protocol. Under the standard sampling protocol, $PM_{2.5}$ is collected at staggered 3-hour intervals over a 9-day period, generating a 24-hour $PM_{2.5}$ sample covering a full diel cycle. Under the MAIA sampling protocol, $PM_{2.5}$ is collected continuously for 24 hours from 9 am to 9 am at a mission-defined frequency, which has been typically every 3 days during the sampling periods used here. The starting dates for MAIA sites are listed in Table A1. SPARTAN samples are analyzed for fine mineral dust concentrations using X-ray Fluorescence (XRF) and a global mineral dust equation (Equation (A1); Liu et al., 2022) including correction of attenuation effects due to mass loading. The 5-year averaged surface fine dust concentrations from SPARTAN sites are listed in Table A1. We use data from sites with at least 10 samples for the 5-year (2019–2023) period after the network began using XRF with samples. A sensitivity analysis requiring at least 50 samples per site is also conducted. This study used 2,296 filters from 25 SPARTAN sites for a total of 10,072 observational days."

We also add to Table A1 the number of days in which a SPARTAN sample was taken. All sites had samples taken on at least 100 days, for a total of 10,072 days with samples, thus affirming the robustness of the measurements and statistics. AERONET also has a large number of measurements at each site. We add at line 123-124:

"The median number of days with AERONET measurements is 168 days for each site."

Comparisons against other dust observations such as IMPROVE are included for ground-based measurements of $PM_{2.5}$ dust over North America in Figure A7, and PM10 dust globally in Figures A8 and A9. We additionally call for more observations over global dusty regions in the conclusion section of line 680-685:

"…we average the multi-year observational data from the CALIOP extinction profile and SPARTAN measured surface dust concentrations for long-term representativeness. This approach benefits from the weak interannual variability of annual mean mineral dust concentrations (Li et al., 2017; Song et al., 2021). Nonetheless, additional observational data will enable further evaluation of the performance of mineral dust simulations."